# Local impact analysis of climate change on precipitation extremes: are high-resolution climate models needed for realistic simulations?

Hossein Tabari[1], Rozemien De Troch[2,5], Olivier Giot[2,6], Rafiq Hamdi[2,5], Piet Termonia[2,5], Sajjad Saeed[3], Erwan Brisson[3], Nicole Van Lipzig[3] and Patrick Willems[1,4]

[1]Hydraulics Division, Department of Civil Engineering, KU Leuven, Kasteelpark Arenberg 40, BE-3001 Leuven, Belgium.
[2]Royal Meteorological Institute of Belgium, Brussels, Belgium.
[3]Department of Earth and Environmental Sciences, KU Leuven, Leuven, Belgium.
[4]Department of Hydrology and Hydraulic Engineering, Vrije Universiteit Brussel, Belgium.
[5]Department of Physics and Astronomy, Ghent University, Belgium.
[6]Plant and Vegetation Ecology, University of Antwerp, Belgium.

*Correspondence to*: Hossein Tabari (hossein.tabari@kuleuven.be: tabari.ho@gmail.com)

**Abstract.** This study explores whether climate models with higher spatial resolution provide higher accuracy for precipitation simulations and/or different climate change signals. The outputs from two convection-permitting climate models (ALARO and CCLM) with a spatial resolution of 3-4 km are compared with those from the coarse scale driving models or reanalysis data for simulating/projecting daily and sub-daily precipitation quantiles. Validation of historical design precipitation statistics derived from intensity–duration–frequency (IDF) curves shows a better match of the convection-permitting model results with the observations-based IDF statistics compared to the driving GCMs and reanalysis data. This is the case for simulation of local sub-daily precipitation extremes during the summer season, while the convection-permitting models do not appear to bring added value to simulation of daily precipitation extremes. Results moreover indicate that one has to be careful in assuming spatial scale independency of climate change signals for the delta change downscaling method, as high-resolution models may show larger changes in extreme precipitation. These larger changes appear to be dependent on the time scale, since such intensification is not observed for daily time scale for both the ALARO and CCLM models.

## 1 Introduction

It becomes evident that climate change will increase the frequency and intensity of extreme events (IPCC, 2007, 2013). Therefore, the impacts of climate change on hydrological extremes such as heavy precipitation events have to be considered when designing and optimizing water infrastructures. The future projection of climate change impact on precipitation usually relies on the simulation results of General Circulation Models (GCMs). However, these results need to be validated against historical precipitation observations prior to any use for local impact studies of climate change. When GCM results are validated based on observations, sometimes large biases are observed especially for extreme precipitation values (van Pelt et al., 2012; van Haren et al., 2013; Tabari et al., 2015), imposing an uncertainty to the GCM projections for the future. The biases in the coarse-resolution GCMs come

from the fact that they disregard some governing features of precipitation at local scale, next to the scale differences when comparing GCM results with local observations (Maraun et al., 2010; Willems et al., 2012). Some previous studies that attempted to assess GCM skill as a function of resolution showed that the performance of GCMs is independent of their resolution (Johnson et al., 2011; Masson and Knutti, 2011). However, given that deep convective phenomena are sufficiently resolved only at spatial resolutions up to less than about 4 km, such dynamical downscaling is expected to be one of the solutions for decreasing the systematic biases and narrowing the gap between GCM outputs and needs for fine-scale precipitation in hydrological and water engineering studies.

One of the methods to dynamically downscale GCM outputs is to drive a Regional Climate Model (RCM) using GCM as initial and boundary conditions. RCMs usually provide an improved description of surface features (topographical, land cover, etc.) and more complex description of atmospheric processes compared to GCMs. This often results in more realistic representation of precipitation variability and of climate feedback mechanisms (IPCC, 2001; Mearns et al., 2004; Christensen and Christensen, 2007; Mayer et al., 2015). Whatever climate models are used, verification of their results under the current climate is needed, because some high-resolution RCMs fail to adequately describe local-scale surface processes (especially in inhomogeneous regions with complex topography) due to the convective parameterization scheme or the characteristics of the GCM they are nested in (Hohenegger et al., 2008; Willems et al., 2012).

High-resolution (convection-permitting resolutions) climate models are of great added value to simulate large convective storms and mesoscale organization (Kendon et al., 2014; Prein et al., 2015). At these resolutions, deep convection is partly resolved and does not need to rely entirely on parameterizations. The representation of the daily cycle in precipitation, extreme events and spatial variability strongly improves for convection-permitting models (Kendon et al., 2012; Prein et al., 2013a, 2013b, 2015; Brisson et al., 2015; Ban et al., 2014, 2015, Fosser et al., 2015, 2016). However, their long-term simulation is restricted due to high computational costs. They are consequently mainly applied for numerical weather prediction (Done et al., 2004; Baldauf et al., 2011; Tang et al., 2013). First simulations for decadal time periods using convection-permitting models point to a stronger increase in extremes compared to coarser resolution integration, but the number of climate change impact studies with these models is limited so far (Hohenegger et al., 2008; Kendon et al., 2012, 2014; Prein et al., 2015).

The use of regional climate models for local impact studies of climate change on precipitation (totals or extremes) has been increased in recent years (e.g. Willems and Vrac, 2011; Olsson et al., 2012; Mearns et al., 2013; Rajczak et al., 2013; Olsson et al., 2015). Nevertheless, in some studies, climate scenarios have been based on a broad set of coarse-resolution GCM results (Deng et al., 2013; Rana et al., 2014; Sun et al., 2015). Now, the question is whether high-resolution climate models truly improve extreme precipitation simulations, and if so, to what extent. This study intends to answer this research question by comparing high-resolution models (RCMs with resolutions between 40 and 3 km) with their driving GCM or reanalysis data for simulating sub-daily and daily precipitation quantiles. Further comparisons are performed for simulating design precipitation statistics derived from intensity–duration–frequency (IDF) curves.

Second research question considered, in case the high resolution climate models show improved extreme precipitation results, is whether this improvement in absolute precipitation values also significantly changes the

relative climate change signal. Hydrological applications of climate change impact analysis often assume that the precipitation change factors, defined as the relative change from historical to future climate conditions, can be obtained from GCM or RCM simulations and applied for impact analysis at finer spatial scales. This is the case for any delta change or perturbation based statistical downscaling method (e.g. Ntegeka et al., 2014; Sunyer et al., 2015). In this study, the validity of this hypothesis is investigated by comparing the climate change signals between the high and coarse scale resolution models. Central Belgium is considered as the study location.

## 2 Climate models

### 2.1 ALARO model

The ALARO-0 model is a high-resolution regional climate model developed by the Royal Meteorological Institute (RMI) of Belgium based on the numerical weather prediction model called Aire Limitee Adaptation Dynamique Developpement International (ALADIN). Hereafter, ALARO is used as shorthand name for the ALARO-0 model described in De Troch et al. (2013). The ALADIN model is the limited area model (LAM) version of the Action de Recherche Petite Echelle Grande Echelle Integrated Forecast System (ARPEGE-IFS). The physics parameterization package of the ALARO model was designed specifically for running at resolutions between 3 and 8 km. The specific characteristics of the Modular Multiscale Microphysics and Transport (3MT) convection scheme used in the ALARO model lead to a good multiscale performance, particularly in convection-permitting resolutions (De Troch et al., 2013). The ALARO simulations for the present climate conditions over Belgium were performed for the periods 1961-1990 and 1981-2010 at resolutions ranging from 40 km down to 4 km, both using a set of simulations forced with ERA-40 or ERA-Interim reanalysis as well as with the CNRM-CM3 GCM for the historical control run (Table 1). For the future climate projections (2071–2100), the CNRM-CM3 GCM under the A1B scenario was used to force the ALARO model (Hamdi et al., 2014).

### 2.2 CCLM model

The other high-resolution climate model used in this study is the COSMO-CLM (CCLM) model. The CCLM is a non-hydrostatic limited area climate model developed by the climate limited-area modeling (CLM) community. The CCLM model is based on the COSMO model (Steppeler et al., 2003), designed by the Deutsche Wetterdienst (DWD) for operational weather prediction. In order to perform climate simulations with the COSMO model, the CLM community provided extensions such as dynamic surface boundaries, a more complex soil model and the possibility to use various $CO_2$ concentration values (Böhm et al., 2006; Rockel et al., 2008).

The model settings are based on a previous study by Brisson et al. (2015), which provide recommendations for performing climate simulations at convection permitting scale. The one-moment microphysical parameterization includes a representation of graupel hydrometeors. In addition, the domain size of this simulation (192x175 gridpoints) is large enough to ensure that the analysis is not affected by the spatial spin-up described in Brisson et al. (2015). The integration scale of global models largely differs from convection permitting scale. A multiple nesting strategy was therefore selected to carry out such simulations (Brisson et al. 2015, 2016). A three-step nesting

strategy was applied with the driving data, either from ERA-Interim reanalysis data or the EC-EARTH GCM, forcing a CCLM at 25 km grid mesh size, which in turn forces a CCLM at 7 km grid mesh size, and next at the final 2.8 km grid mesh size. Model simulations were performed for the period 2001-2010, and a thorough evaluation of the statistics of precipitation, temperature and cloud characteristics was recently performed (Brisson et al., 2016). The CCLM driven by EC-EARTH was performed for the period 2000-2010 and 2060-2069 using the RCP4.5 emission scenario (Table 1). Hereafter, the driving GCM or reanalysis dataset is shown as subscript to the name of the RCM. As the control run of the EC-EARTH GCM ends in 2009, its data for the period 2000-2009 were used for comparing with the driven CCLM simulations.

## 3 Methodology

In this study, simulations of sub-daily and daily precipitation quantiles from the climate models are analyzed. For the future climate analysis, the climate change signals are obtained as relative changes of precipitation intensities calculated as the ratios of precipitation quantiles derived from each climate model scenario simulation over those from the corresponding climate model control simulation with same non-exceedance probability or return period. This methodology has been applied in several recent climate change studies, e.g. on the basis of statistical downscaling applying quantile mapping or quantile perturbations (Willems and Vrac, 2011; Gudmundsson et al., 2012; Maraun, 2013; Ntegeka et al., 2014; Rana et al., 2014; Sunyer et al., 2015) and also a similar procedure for analyzing decadal precipitation anomaly (Tabari et al., 2014; Tabari and Willems, 2016). For sub-daily precipitation, independent extremes are selected using a Peak Over Threshold (POT) method. The POT selection is done based on three criteria for inter-event time, inter-event low precipitation and peak height, similar to those presented by Willems (2009) for extracting POT values for discharge. The inter-event time is the main criterion for extraction of POT values. Following Willems (2013), an inter-event time of 12 hours is selected, implying that two successive precipitation peaks within the same day or night are considered as one extreme event. In other words, two consecutive precipitation extremes are interpreted to be independent based on this criterion when the time between the two events exceeds 12 hours. Extreme precipitation is defined in this study as precipitation with return period (T) higher than 1 year. The return period is in this study calculated in two different ways: empirically based on the rank of the extracted POT values (n/i, where n and i are the length of the study period and rank, respectively; i = 1 for the highest value); and theoretically after calibrating an extreme value distribution to these POT precipitation extremes. Also for the calculation of the precipitation change factors for given return periods, these two different approaches were followed and compared: empirical data based and extreme value distribution based change factors. For the distribution based change factors, first a distribution is fitted separately to the extreme values of the control and scenario runs of the climate models. Afterwards, change factors are computed as a ratio between the fitted distribution values of the scenario and control runs.

In addition to the quantile analysis, the historical simulations of the climate models are validated based on precipitation intensity–duration–frequency (IDF) curves which are typically used for design storm calculations and related designs, e.g., urban drainage systems and hydraulic structures. The IDF curves for 1-month, 1-year and 10-year return periods and for durations from 10-15 minutes up to one month are developed for the control runs of the

climate models as well as the observations. The IDF curves are derived based on POT extreme value statistics after
calibration of two-component exponential distributions, following Willems (2000). In this paper, the precipitation
intensities of given return periods are referred to as design precipitation quantiles.
For the climate models, precipitation data are extracted from a matrix of 3×3 model grid points (9 cells)
surrounding the closest model grid point to Uccle station in Central Belgium. This station is selected because it has
high quality 10-min observations recorded with same instrument since 1898 (Demarée, 2003). In addition to the 10-
min station observations, daily E-OBS gridded data (v12.0, Haylock et al., 2008) for 27.8 km and 55.7 km are used.
These gridded data are aggregated to larger pixels of 167 km and 334 km to be consistent with the grid mesh size of
the driving GCMs and reanalysis data. The aggregation is also performed to upscale the outputs of the convection-
permitting climate models to check the accuracy of the spatial structure in the models.
**4 Validation of precipitation simulations**
The capability of the climate models to simulate the present-day precipitation is evaluated before investigating
future precipitation changes. Prior to this performance evaluation, the precipitation extremes from the model grid
cell covering Uccle station are compared with those from neighboring cells for possible outlier or unrealistic values.
The analysis shows spatial consistency in the frequency of daily and sub-daily precipitation extremes for both the
ALARO and CCLM models. As an example, Fig. 1 illustrates hourly precipitation extremes in a matrix of 3×3
ALARO$_{ERA-Interim}$ 4 km model grid points surrounding the closest model grid point to Uccle station for summer and
winter seasons. It is seen that hourly precipitation extremes in gridcell 5 covering Uccle station are consistent with
the ones in the neighboring gridcells. Another preliminary analysis is performed to compare point and pixel
interpolated Uccle precipitation observations, which are used as reference for the model performance evaluation
(Fig. 2). The comparison is done for the periods 1961-1990 and 2001-2010, which are the control periods of the
ALARO and CCLM models, respectively. The precipitation extremes from the pixel E-OBS data follow the pattern
of the point observations and the extremes are well represented in the pixel dataset. The smaller amounts from the
gridded dataset is due to the fact that spatial averaging smooths out the extreme values (Hofstra et al., 2010; Sunyer
et al., 2013).
The validation results of the daily precipitation quantiles simulated by the ALARO convection-permitting
models and its boundary conditions based on the point and pixel interpolated Uccle observations for the summer
season (June-July-August: JJA) are shown in Fig. 3. The precipitation extremes for each model run are evaluated on
the native model grids, and are then aggregated to a larger model grid size in order to ensure a fair comparison. For
the aggregation purpose, the coarsest grid is used as reference. It means that, for instance for the ALARO model, the
evaluation of the model with 4 and 10 km resolutions is carried out on the coarser 40 km grid. The results on the
native model grids are presented to evaluate whether the available climate model runs are of direct use for climate
change impact analysis in urban hydrology. The native daily precipitation extremes reveal the largest extreme values
for the ALARO$_{ERA40}$ 4 km model (Fig. 3a). However, this might be due to the precipitation decrease after the spatial
averaging. The overestimation of the ALARO runs nested in the ERA40 reanalysis data is also evident on the native
model grids, while the extreme simulations of the ALARO$_{CNRM-CM3}$ model with 4 km resolution are in between the

point observations and the gridded ones with a grid size of 27.8 km, which shows good accuracy of these simulations. When comparing the model results at the same grid size (Fig. 3b), the ALARO$_{ERA40}$ 40 km outputs are larger than those from the ALARO$_{ERA40}$ model for the higher resolutions at 4 and 10 km. This indicates the role of spatial scale in the climate modeling by the ALARO model driven by the ERA40 reanalysis data. Also other authors reported no improvements in the simulations of daily mean precipitation by the convection-permitting models compared with large scale climate models (Chan et al., 2013; Fosser et al., 2015). Some other researchers found improvements especially over mountainous areas (Prein et al., 2013b; Ban et al., 2014), implying region and model dependency for simulation of daily mean precipitation. In our study, the higher skill of the ALARO$_{CNRM-CM3}$ model in simulation of summer precipitation extremes appears to be because of a better representation of the small-scale characteristics and spatial variability relevant for convection (Fig. 3b). The CNRM-CM3 GCM and ERA40 reanalysis data used as the boundary conditions of the ALARO model show a systematic underestimation especially for the higher return periods (Fig. 3a). The convection parameterization has been found to be responsible for this underestimation (Kendon et al., 2014).

As for the CCLM model, the native daily precipitation quantiles from the 2.8 km runs are larger for most of the cases (Fig. 4a). After upscaling of the finer resolution models (2.8 and 7 km) to the larger scale (25 km), the results of the models become similar (Fig. 4b). The driving EC-EARTH GCM and ERA-Interim reanalysis underestimate the summer extremes, probably due to the misrepresentation of the convective processes. When the results of the driven GCM and reanalysis data are compared with the ones of the CCLM, the larger and more accurate simulations of the CCLM model is observed for summer when convection becomes dominant. This confirms the finding that higher resolution results in more extreme precipitation in climate models (Jacob et al., 2014). The increasing skill of RCMs with increasing model resolution for simulation of the spatio-temporal characteristics of summer precipitation has been found by using the high-resolution models, although limited in application (Rauscher et al., 2010; Kendon et al., 2012). Nevertheless, a comparison between the CCLM outputs of different resolutions does not show a clear difference, neither in precipitation intensity or in simulation skill (Fig. 4b).

The extreme precipitation (averaged over the extreme events with T > 1 year) simulations of the climate models versus spatial scale for both summer and winter seasons are shown in Fig. 5. Taking the spatial scale difference into account and averaging the extreme values with T > 1 year, the ALARO$_{ERA40}$ simulations are closer to the observations compared with the ALARO$_{CNRM-CM3}$ model. Decease in systematic biases in the large scale climate in reanalysis-driven RCM simulations was also reported by Maraun et al. (2010). They also pointed out that these RCMs are capable of reproducing the actual day-to-day sequence of weather events. The good accuracy of the CCLM model, large underestimations of CNRM-CM3 and EC-EARTH, slight overestimation of ERA-Interim data and slight underestimation of ERA40 data for summer precipitation extremes are also obvious from these plots. As expected, the percentage bias of the climate models (not shown) decreases as the time scales get larger (i.e., weekly and monthly).

The validation of the climate model simulations for the summer season in terms of IDF statistics is shown in Fig. 6 for time scales in the range between 10-15 minutes and 30 days. The IDF curves are plotted with reference to design precipitation intensities from the station and E-OBS pixel data over the Uccle location (Central Belgium).

Comparing the hourly simulations of the $ALARO_{ERA40}$ model with different resolutions shows the greater intensities for finer resolutions. In terms of accuracy, all of the ALARO runs except the $ALARO_{CNRM-CM3}$ for 10-year return period and the $ALARO_{ERA40}$ 40 km for both return periods underestimate the station observations and overestimate the gridded observations (extrapolated for sub-daily precipitation based on extreme value distribution). Regarding 3- and 6-hourly time scales, the ALARO model simulates more intense precipitation of 10-year return period in comparison to both the station and gridded observations. The model underestimates (overestimates) extreme precipitation of 1-year return period and 3- and 6-hourly durations when compared with the station (gridded) observations. Daily precipitation intensity of 10-year return period derived from the point observations is underestimated by the $ALARO_{ERA40}$ and $ALARO_{ERA-Interim}$ runs and overestimated by the $ALARO_{CNRM-CM3}$ run, while all the runs overestimate the pixel observations-based statistics. All the ALARO runs except the $ALARO_{ERA-Interim}$ simulate larger daily precipitation extremes of 1-year return period. A comparison between the ALARO 4 km runs nested in reanalysis data for larger time scales between 5 and 30 days shows overestimation of the $ALARO_{ERA40}$ and underestimation of the $ALARO_{ERA-Interim}$ with respect to the station data, whereas both of them overestimate the pixel observations-based statistics. The other ALARO 4 km run ($ALARO_{CNRM-CM3}$) underestimates both the point and pixel observations-based statistics for these larger aggregation levels (5, 10, 15 and 30 days).

The CCLM model simulates less intense 15-min precipitation of 10-year return period (Fig. 6). However, this underestimation changes to overestimation for larger sub-daily aggregation levels. For the sub-daily design storms of 1-year return period, the CCLM model generally underestimates the station observations, while both over- and underestimations are seen in comparison with the gridded observations. However, the EC-EARTH GCM extremely underestimates both the gridded and raingauge observations for the 10-year return period. This supports the recent findings for underestimation of heavy hourly precipitation during summer by large scale climate models and more accurate simulations of convection-permitting models (Chan et al., 2013, 2014; Ban et al., 2014; Fosser et al., 2015). In the case of daily duration, which are less important for urban drainage applications, the CCLM runs underestimate (overestimate) precipitation intensity of 1-year return period in comparison with the point (gridded) observations (Fig. 6). The underestimation of higher intensities by the CCLM 2.8 km run for summer has also been reported in the literature (Fosser, 2014). For the daily precipitation extremes of 10-year return period, the 2.8 km runs and the $CCLM_{EC-EARTH}$ 25 km underestimate (overestimate) precipitation intensity from the point (gridded) observations, while the rest of the CCLM runs show the opposite behavior. For the larger aggregation levels between 5 and 30 days, the precipitation intensities of 1-year return period derived from both the point and pixel observations are underestimated by all the CCLM runs. For the 5-day duration and 10-year return period, underestimation of the station observations-based statistics and overestimation of the pixel observations-based statistics are seen for all the CCLM runs except for the 7 km runs. The $CCLM_{ERA-Interim}$ 2.8 and 7 km runs simulate larger precipitation extremes for the 10-, 15- and 30-day durations of 10-year return period, whereas the $CCLM_{ERA-Interim}$ 25 km run simulates smaller extremes. The similarity between the CCLM 2.8 and 7 km runs is expected to be explained by the similarity in lateral boundary conditions since the CCLM 2.8 km model is nested in the CCLM 7 km model. However, the difference between these runs becomes obvious when the convection is dominant in sub-daily summer precipitation as they treat deep convection in different ways. The $CCLM_{EC-EARTH}$ 25 km run shows the

same pattern as the $CCLM_{ERA-Interim}$ run: underestimation of extreme precipitation intensity for the 10-, 15- and 30-day durations of 10-year return period. Both over- and underestimations are seen for the $CCLM_{EC-EARTH}$ 2.8 and 7 km runs for the 10-, 15- and 30-day durations of 10-year return period (Fig. 6).

For the winter season (December-January-February: DJF), the results show overestimations of the ALARO and CCLM models (Fig. 5). As winter precipitation over Belgium is mainly controlled by large scale circulation, an improvement in the simulations of convection-permitting models in comparison to the parent large scale models is less expected for the winter season. Although improved simulations of winter precipitation by convection-permitting model have been reported for regions with complex topography (Ikeda et al., 2010; Rasmussen et al., 2011) due to better resolved orography (Prein et al., 2015), this effect is less relevant for Belgium which is more flat. Whereas winter daily precipitation extremes are systematically overestimated by the ALARO model, the driving CNRM-CM3 GCM slightly underestimates the winter extremes (Fig. 5). Deficiency of very high resolution climate models in simulation of winter precipitation extremes is because the fronts and synoptic depressions that cause the dynamical processes driving winter precipitation events have scales of $10^2$-$10^3$ km. This deficiency has been demonstrated by Hong and Leetmaa (1999) and Chan et al. (2013). For the CCLM model, when the $CCLM_{EC-EARTH}$ 2.8 and 7 km simulations are compared with those of the $CCLM_{ERA-Interim}$ 2.8 and 7 km for the daily winter extremes, the overestimations of the earlier runs are higher than the later ones, while for larger time scales (weekly and monthly) the opposite pattern is observed.

## 5 Future precipitation changes

To cope with the scale difference and the biases shown in the previous section, state-of-the-art climate change impact analysis makes use of statistical downscaling. One of the popular downscaling methods is the delta change method. Different versions exist for that method: from the simple basic method to more advanced methods such as the quantile perturbation method. In this type of methods, the intrinsic assumption is made that the bias under future climate conditions is identical to the bias in current climate conditions. This is implemented through the use of "change factors" applied for historical precipitation quantiles. Another important assumption that is made by these methods is that the change factors are spatial scale independent, such that the scale difference, although it is an issue for the absolute precipitation intensity values, is less an issue for the delta change methods at which relative changes are applied. The latter assumption is tested next. In this context, the relative changes in precipitation quantiles between the future and historical simulations of climate model runs were calculated to compare the convection-permitting models and their driving GCMs. These change factors were computed for winter and summer seasons as sub-daily and daily precipitation quantiles from the scenario period divided by those from the control period with the same return period (change factor equal to one means no change).

The change factors derived from the empirical data, and the ones after use of the extreme value distribution in precipitation extremes for winter and summer seasons computed by the $ALARO_{CNRM-CM3}$ model and the driving CNRM-CM3 GCM are shown in Fig. 7. From a comparison between the empirical data based change factors and those based on the extreme value distributions, it is seen that the extreme value distribution fitting smooths out abrupt changes and random variations in the change factors, making the results easier to interpret. In fact, the

distribution fitting removes the randomness involved in the high return periods of the empirical data for summer,
leading to a slight difference in the range of changes. However, for the winter season the change factors from the
two methods have similar ranges. The change factors obtained from the extreme value distribution fitting are further
discussed here. The ALARO$_{CNRM-CM3}$ projects an increasing signal in the range of 26% to 69% for daily winter
extremes. The projected increase is even higher for hourly winter extremes, ranging between 37% and 120%. When
the change factors computed by the ALARO$_{CNRM-CM3}$ are compared with those obtained from the driving CNRM-
CM3 GCM, more or less the same conclusion can be made: an increasing signal for daily winter extremes between
23% and 67%. For the summer season, the change factors from the ALARO$_{CNRM-CM3}$ model and the parent CNRM-
CM3 GCM are around one, meaning no change in daily summer extremes. However, smaller hourly summer
extremes are expected based on the ALARO$_{CNRM-CM3}$ model projections with a decreasing signal down to -26%.
Generally, it can be inferred from the results that, at synoptic (daily) scale, the projections by the ALARO model are
consistent with those from the driving GCMs. De Troch et al. (2013) pointed out that an increase in spatial
resolution in the ALARO model is not as important as the parameterization scheme used for extreme precipitation
modeling at the daily scale.
Fig. 8 shows the change factors for daily and 3-hourly precipitation computed using the CCLM$_{EC-EARTH}$ model
with different spatial resolutions and the driving EC-EARTH GCM for winter and summer seasons. The change
factors for all extreme events with T > 1 year are shown in this figure. For the winter season, the change factors for
both daily and 3-hourly precipitation decrease as the model's resolution increases. Nevertheless, the change factors
for all the CCLM runs are higher than those for the driving EC-EARTH GCM. A larger change is projected for 3-
hourly precipitation compared with daily precipitation. For summer, the greatest change is obtained for 3-hourly
precipitation extremes from the CCLM$_{EC-EARTH}$ 2.8 km run. This increasing signal goes as high as 55%. When the
change factors in 3-hourly precipitation extremes from the CCLM$_{EC-EARTH}$ runs are compared with those from the
driving EC-EARTH GCM, the results show an amplification of the future climate change signals by the CCLM
model: maximum changes of 55%, 11% and 14% respectively for 2.8, 7 and 25 km runs versus a maximum change
of 8% for the driving EC-EARTH GCM. This amplification is not evident for the daily scale. Intensification of
change in sub-daily precipitation extremes that are not simulated by large scale models was also found by Kendon et
al. (2014). The results also reveal that sub-daily precipitation extremes during summer are expected to change at a
higher rate compared to daily extremes. Generally, it can be inferred that there is an increase in the change factors of
sub-daily precipitation when going from parameterized convection to the convection-permitting scale.
**6 Concluding remarks**
A comparative study between the convection-permitting climate models with a spatial resolution from 2.8 km up to
40 km and driving GCMs or reanalysis data was performed to check whether the models with higher resolution
provide more accurate precipitation simulations. Another analysis was performed to validate the spatial scale
independency assumption of climate change signals for the delta change downscaling method. The results show that
whereas winter daily precipitation extremes are generally overestimated by the ALARO and CCLM models,
improved results for summer precipitation extremes are observed especially for sub-daily time scale. This suggests

the added value of convection-permitting climate models to simulate summer sub-daily extremes because of either better representation of deep convection or more detail of the land surface. The results moreover indicate that the difference between the convection-permitting models and the parent GCMs or reanalysis data decreases as the time scales get larger (i.e., weekly and monthly). Based on the precipitation statistics derived from IDF curves, the ALARO and CCLM models mostly underestimate local sub-daily precipitation, but still better simulate it compared with parent GCM or reanalysis data when available. Higher precipitation intensities by finer resolution models are a result of better representation of small-scale convective precipitation by these models.

To investigate whether or not the climate change signals from the convection-permitting models are more or less the same as those from the large scale driving GCMs, the relative changes were computed for precipitation extremes during summer and winter. For the ALARO model, it can be concluded that, at synoptic (daily) scale, the change factors for the ALARO model are comparable with the ones from the driving CNRM-CM3 GCM. In the case of the CCLM model, the results reveal an intensification of climate change signals for the CCLM model compared with the driving EC-EARTH GCM for the 3-hourly time scale. Comparing change factors for 3-hourly and daily precipitation, a larger change is projected for 3-hourly precipitation for both winter and summer seasons. When the change factors derived from the extreme value distribution are compared with those from the empirical data, it is seen that for both ALARO and CCLM models the climate change signals derived from extreme value distribution fitting are slightly different from the ones obtained from the empirical data for summer due to the removed randomness in the empirical data by the distribution fitting. However, for the winter season the change factors obtained from the two approaches cover more or less the same range.

In summary, because the results of this study indicate that the local sub-daily summer precipitation simulations of the high-resolution climate models are closer to the observations, their future projections are expected to be more accurate than those of the driving GCMs. These climate change signals obtained from the high-resolution models may differ from the ones based on the coarse-resolution models, as a result of improved representation of complex landscape and land surface processes in high-resolution models. However, the resulting precipitation change from these high-resolution climate models should not be interpreted as an exact number because of their limited number. More runs with high-resolution models are required to check the consistency among models. In the same way as an ensemble approach on climate models provides uncertainty estimates on the climate change signals, an ensemble of the high-resolution models provides uncertainty estimates on the difference between the climate change signals of fine versus coarse scale. Also, the statistical significance of the difference in climate change signals at fine versus coarse scale can be tested in such approach. From the comparison in this study, the results of the CCLM$_{EC-EARTH}$ model indicate an increase in the change factors in sub-daily summer extremes when going from parameterized convection to the convection-permitting scale. This amplification is not evident at the daily time scale. For the ALARO model also the higher resolution models show changes in the same range as the coarse resolution models for daily precipitation. The differences appear to be a function of time scale, season and climate model. Different procedures for convection parameterization in the CCLM and ALARO models and different boundary conditions (the first one is nested in the EC-EARTH model from CMIP5 and the later in the CNRM-CM3 model from CMIP3) might explain the discrepancy between the results of the two models. The differences in time scale and season is

expected to be explained by more realistic simulation of the mesoscale processes involved during sub-daily summer precipitation extremes by convection-permitting models. The results also show an amplification of the change from daily to sub-daily precipitation for both ALARO and CCLM models, which casts a doubt on the validity of the temporal scale independency assumption of climate change signals.

*Author contributions.* The simulations of the ALARO climate model were performed in the Royal Meteorological Institute of Belgium (RMI) by R. De Troch, O. Giot, R. Hamdi and P. Termonia. The CCLM climate model was implemented by S. Saeed, E. Brisson and N. Van Lipzig in the Earth and Environmental Sciences Department of KU Leuven. H. Tabari and P. Willems developed the methodology and performed the analyses. The paper was prepared by H. Tabari and P. Willems with substantial contributions from all co-authors.

*Acknowledgements.* This study was partly supported by research projects for the Flemish Environment Agency (Division Operational Water Management and Environmental Reporting), and partly by the Belgian Science Policy Office (CORDEX.be project, BRAIN-be programme) and the European Union's Horizon 2020 research and innovation programme (project BRIGAID, grant agreement No 700699).

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

**Table 1** The convection-permitting model runs used in this study.

| Climate model | Driving GCM/reanalysis | Spatial scale (km) | Temporal scale | Control period | Scenario period | Data coverage |
|---|---|---|---|---|---|---|
| CCLM | ERA-Interim | 2.8 | 15 min | 2001-2010 | - | whole year |
| | ERA-Interim | 7 | hourly | 2001-2010 | - | whole year |
| | ERA-Interim | 25 | 3 hourly | 2001-2010 | - | whole year |
| | EC-EARTH | 2.8 | 15 min[1] | 2001-2010 | 2060-2069 | whole year |
| | EC-EARTH | 7 | hourly | 2001-2010 | 2060-2069 | whole year |
| | EC-EARTH | 25 | 3 hourly | 2001-2010 | 2060-2069 | whole year |
| ALARO | ERA-Interim | 4 | hourly | 1981-2010 | - | whole year |
| | CNRM-CM3 | 4 | hourly | 1961-1990 | 2071-2100 | whole year |
| | ERA40 | 4 | hourly | 1961-1990 | - | summer |
| | ERA40 | 10 | hourly | 1961-1990 | - | summer |
| | ERA40 | 40 | hourly | 1961-1990 | - | summer |

[1] $CCLM_{EC-EARTH}$ data for the scenario period are available for the hourly time scale.

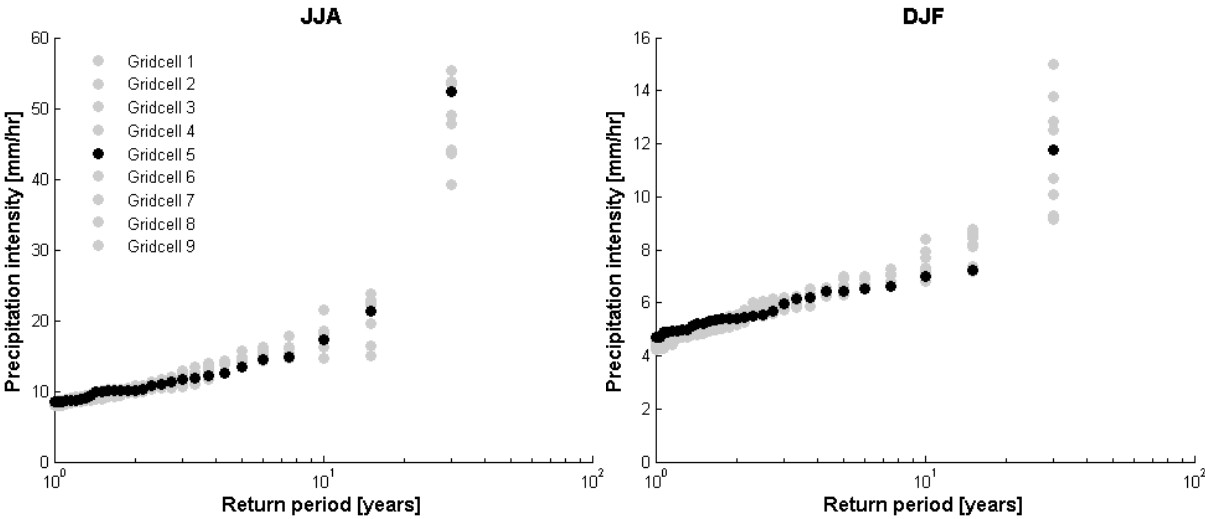

**Figure 1**. Hourly precipitation extremes in a matrix of 3×3 ALARO$_{\text{ERA-Interim}}$ 4 km model grid points surrounding the closest model grid point to Uccle (Gridcell 5), for summer (left) and winter (right) seasons (historical climate: 1961-1990).

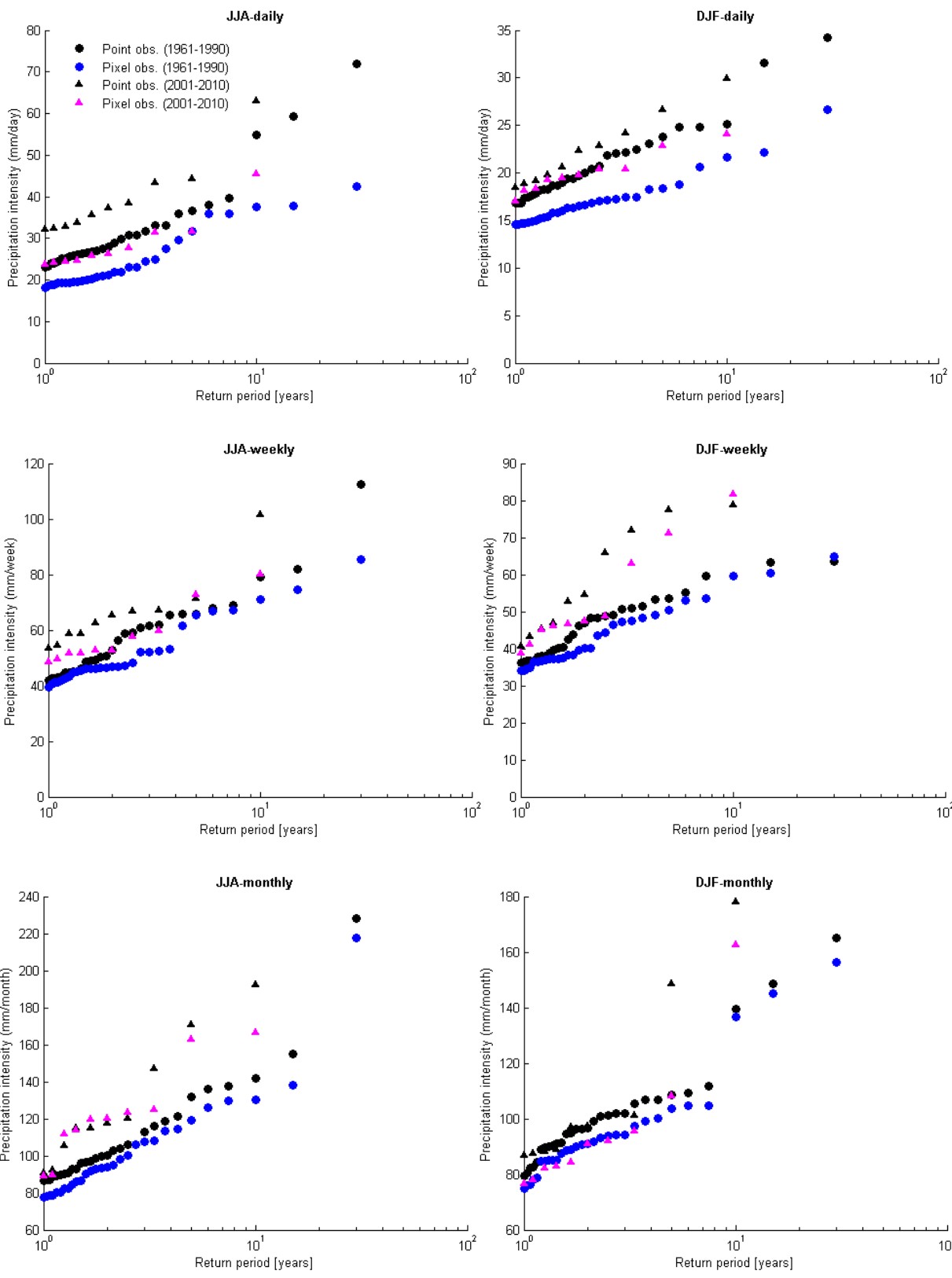

**Figure 2.** Comparison between point and pixel interpolated (spatial resolution of 27.8 km) Uccle precipitation of different time scales for summer (left column) and winter (right column).

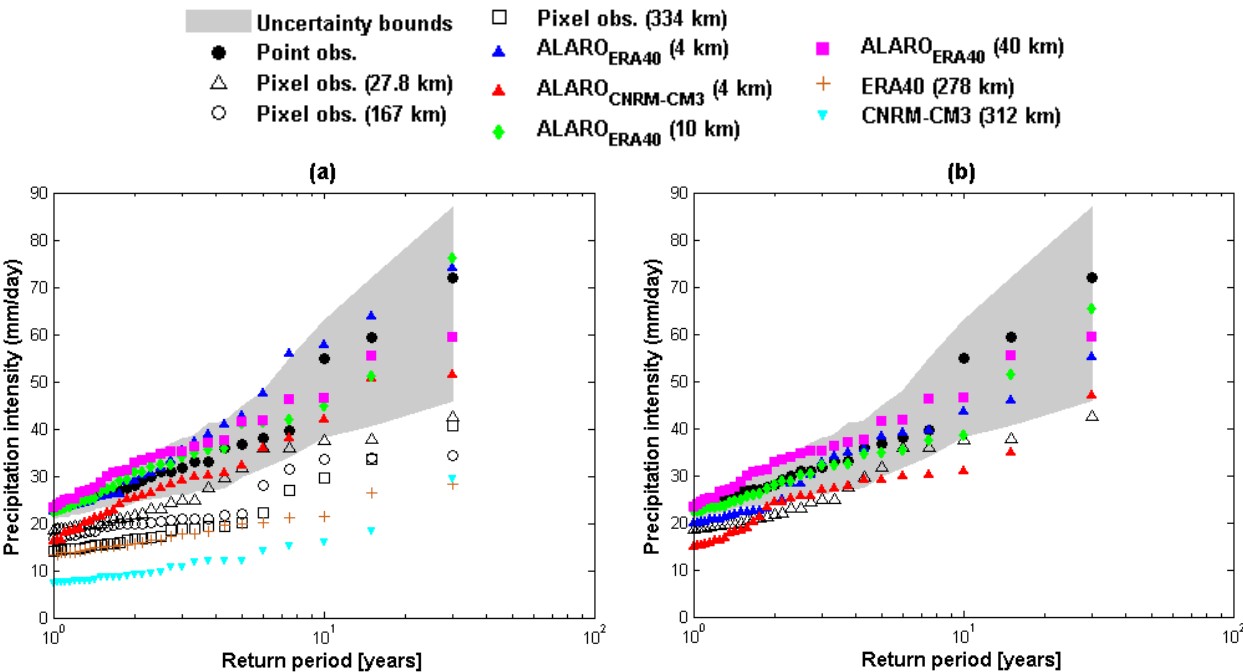

**Figure 3.** Validation of the native (**a**) and aggregated (**b**) daily precipitation quantiles (1961-1990) for the ALARO model and its driving GCM or reanalysis data based on Uccle observations, for summer season (shaded areas show at-site confidence intervals for the point observations using the bootstrap-based 95% confidence intervals).

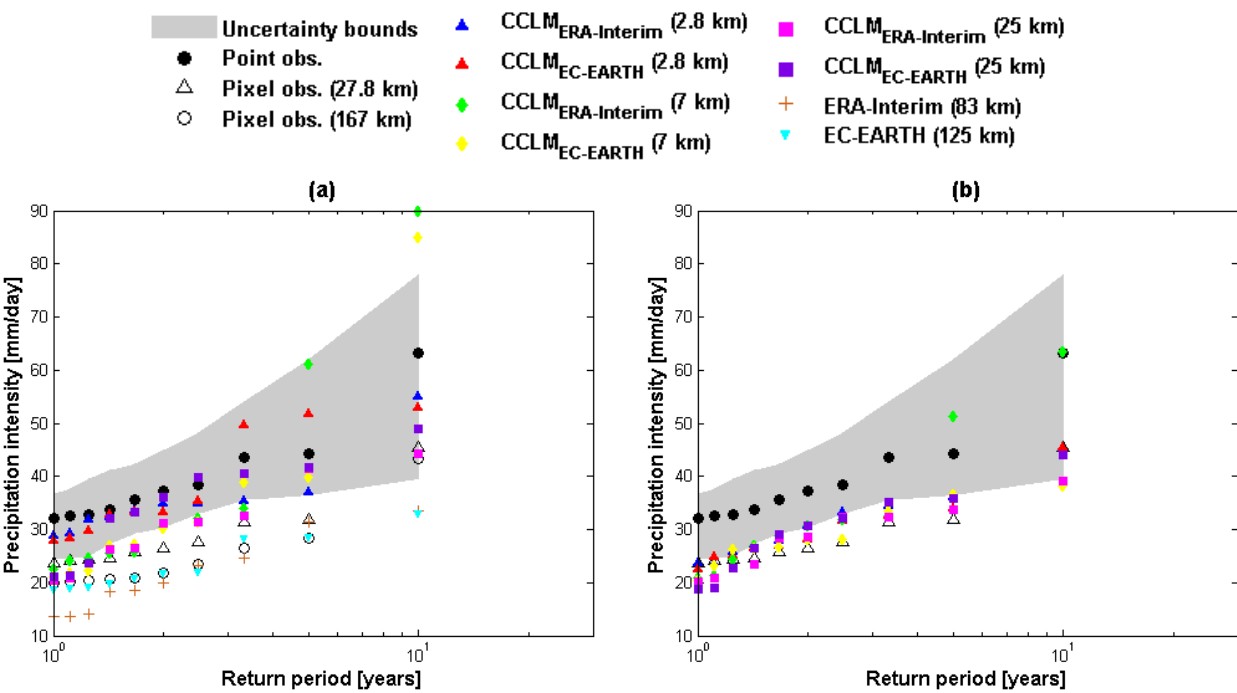

**Figure 4.** Validation of the native (**a**) and aggregated (**b**) daily precipitation quantiles (2001-2010) for the CCLM model and its driving GCM or reanalysis data based on Uccle observations, for summer season (shaded areas show at-site confidence intervals for the point observations using the bootstrap-based 95% confidence intervals).

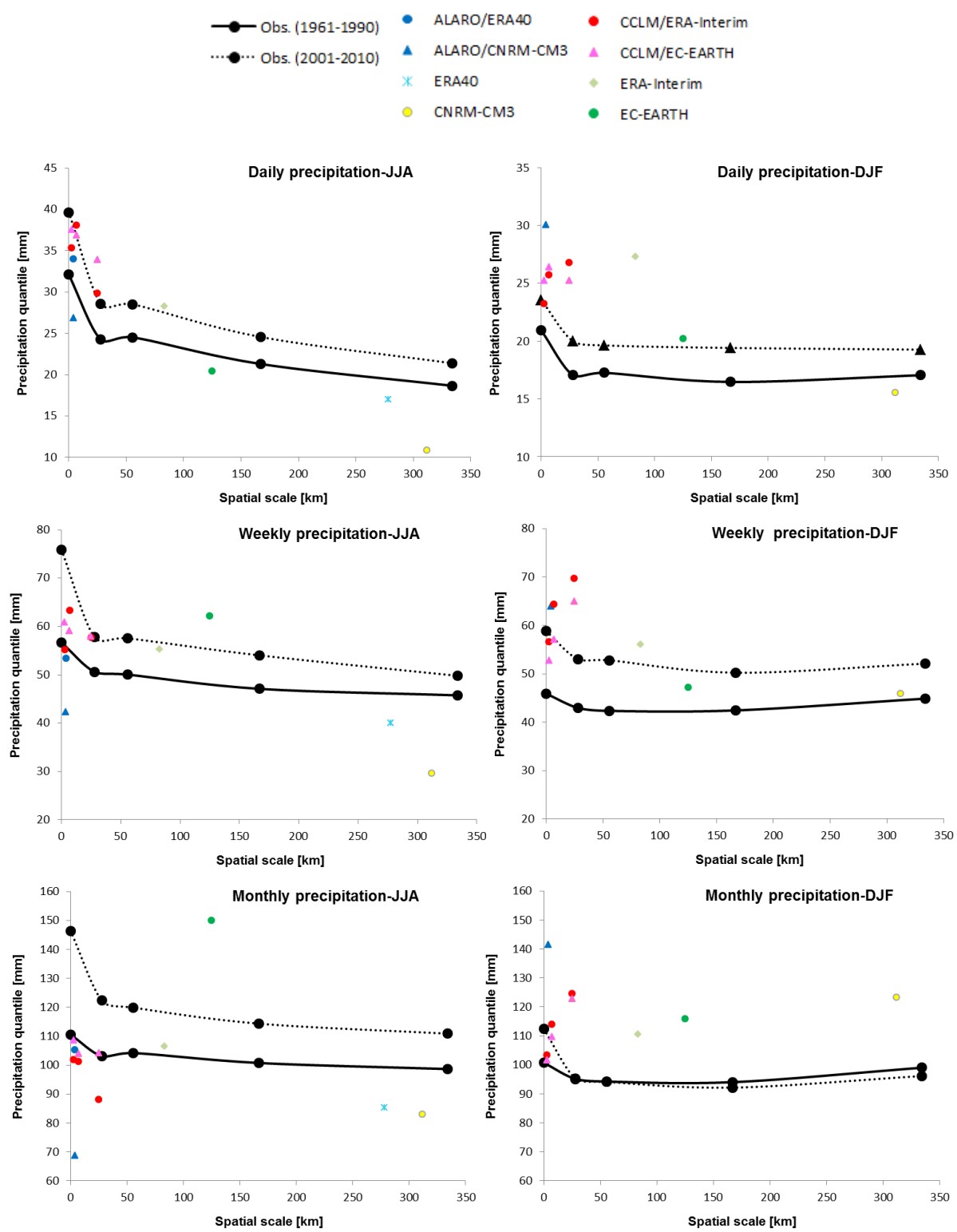

**Figure 5.** Validation of the extreme precipitation (averaged over the extreme events with T > 1 year) simulations for the ALARO, CCLM and the driving GCMs or reanalysis data based on point and pixel interpolated Uccle observations for summer (left) and winter (right) seasons, versus the models' spatial scale.

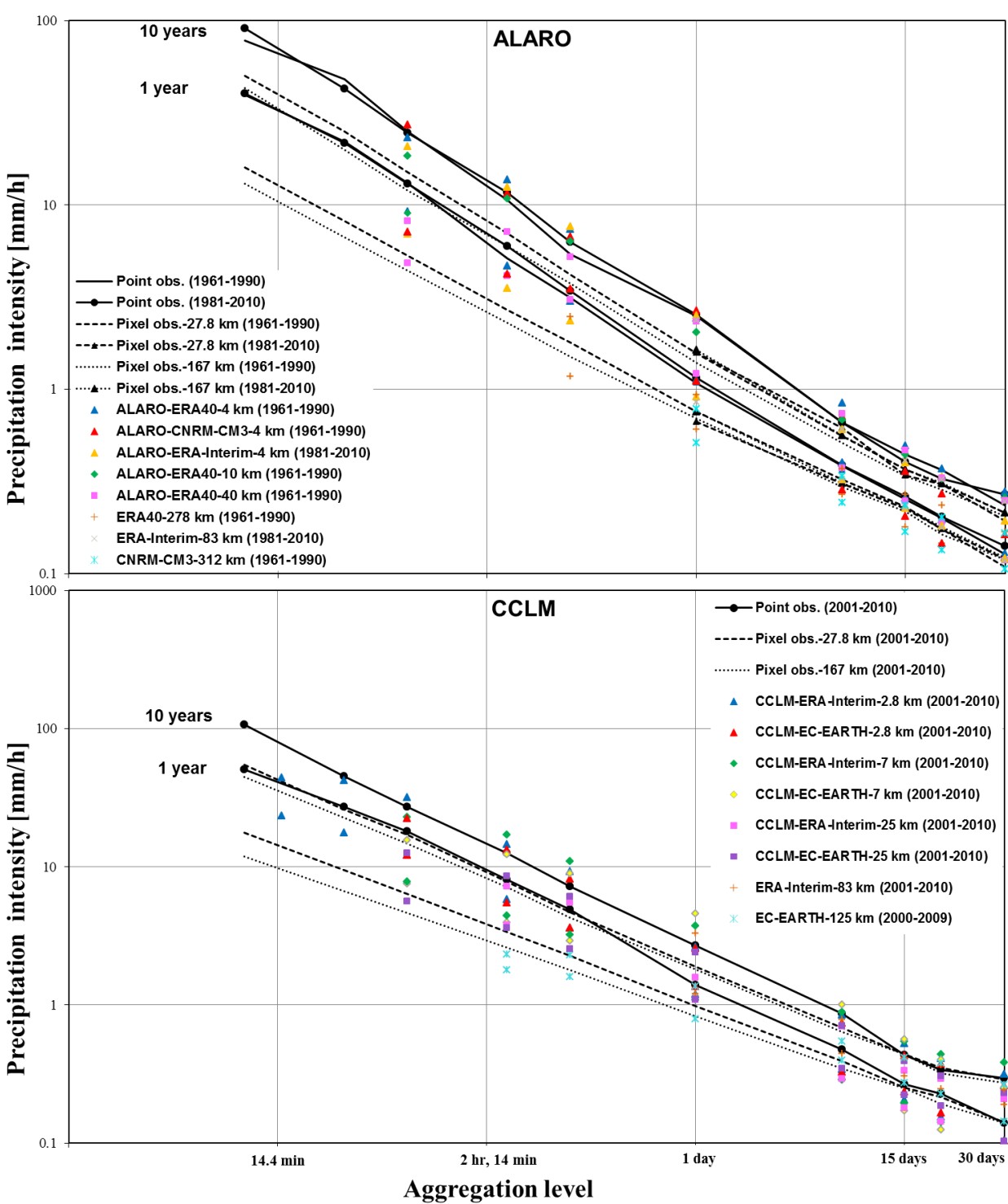

**Figure 6.** Comparison of historical IDF-relationships based on point and pixel interpolated Uccle observations, with the CCLM, ALARO and the driving GCM or reanalysis results for summer season (IDF curves for the E-OBS pixel data were extrapolated for the sub-daily time scales based on extreme value distribution).

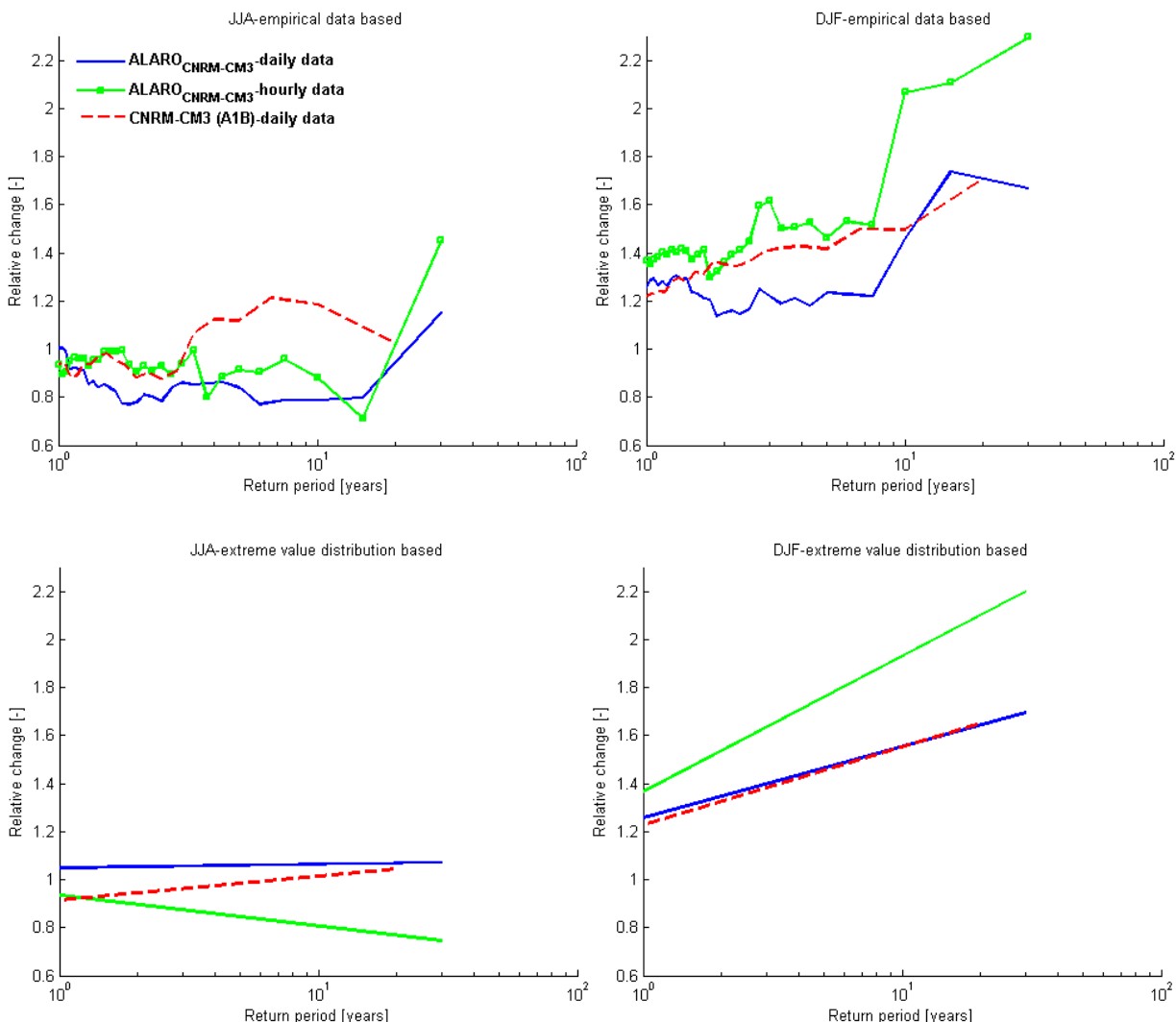

**Figure 7**. Change factors for daily and hourly precipitation quantiles computed using the ALARO<sub>CNRM-CM3</sub> 4 km and the driving CNRM-CM3 (A1B) for summer (left column) and winter (right column) seasons, obtained from the empirical data (top figures) and after use of the extreme value distributions (bottom figures).

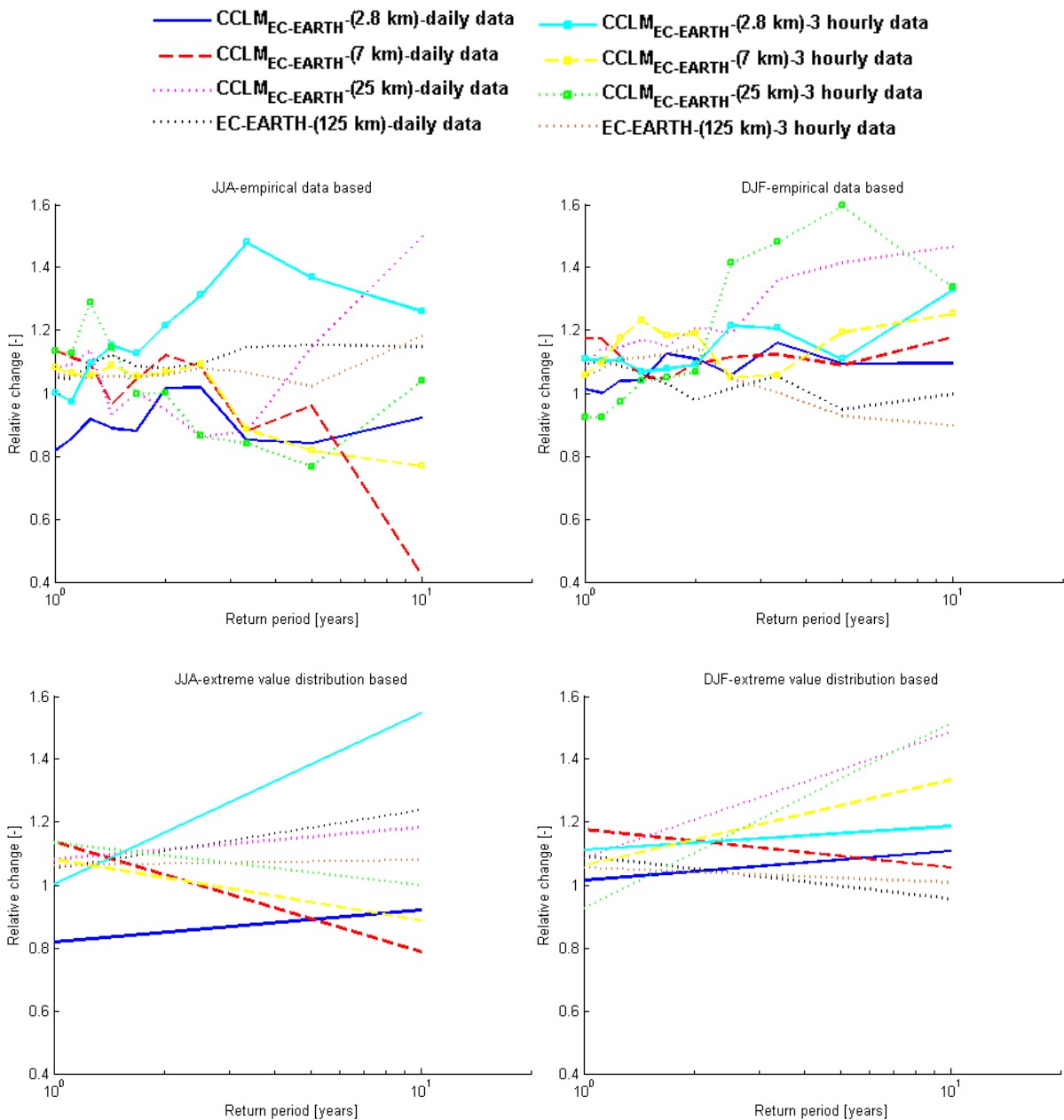

**Figure 8**. Change factors for daily and 3-hourly precipitation quantiles computed using the CCLM$_{EC-EARTH}$ 2.8, 7, 25 km for summer (left column) and winter (right column) seasons, obtained from the empirical data (top figures) and after use of the extreme value distributions (bottom figures).