# Peer review of "Local impact analysis of climate change on precipitation"

_Hydrology and Earth System Sciences, 2016_

## Referee Comment (RC1) · K. Arnbjerg-Nielsen (Referee) · 15 Mar 2016

I accepted to review this paper because it is of interest to me. I have however had collaboration with one of the authors leading to numerous publications in the period 2012-2014. I have checked the HESS guidelines of conflict of interest and do not think there is a conflict. Nevertheless I have been in contact with the editor and will make my review public to avoid any misunderstandings.

Overall the paper is well written and shows that the authors have a comprehensive overview of the relevant literature. Further, the work is a logical extension of the work we have done together. The study fits well within the special issue to which it is submitted. It specifically addresses the gab in knowledge on to which spatio-temporal scale

dynamic downscaling should be performed. So I I think the work is novel and should be published. I have however some comments that I would like the authors to address that I think will lead to an improved paper.

It is probably too late to change now, but I am surprised that you use the A1B scenario for the climate projections. The simulations seem to have been made specifically for this study and hence I had expected an RCP-formulation of the projections.

The CCLM model formulation with three nestings seems very complicated. I have no experience with triple nesting and would like to see some references to work indicating that this is a feasible approach. Also a few words on the approach would be nice. Do you apply some sort of nudging or are you only providing boundary conditions? I would have preferred to go directly from the 25*25 km to the highest resolution and then let computational cost define the area. In any case it is unclear whether the 7*7 km is non-hydrostatic or not. So I would appreciate more information on this crutial step.

On page 5 and 6 you have a quite detailed interpretation of how the various statistics perform. It would be nice to have a metric of uncertainty to distinguish between sampling errors and actual signals from the simulations. The most simple would be to include the at-site confidence intervals for the point observations. Another solution could be to consider the variation in the extreme statistics caused by observed decadal oscillations.

It is not clear to me exactly how the datapoints in the figures are derived. In Figures 1 and 3 it seems that the points are the raw model output statistics (MOS) plotted using a plotting position formula (which one?). This ignores the dependency of extremes on spatial scale often described by an Areal Reduction Factor (ARF). Plotting these values on the same graph implies that the numbers can be compared directly, which is not the case. Please clarify what you do and consider to modify the graphs by incorporating ARFs. In Figure 1 I miss an explanation on how you can show 30 year return periods when using 9 years of observations for the CCLM model (it looks like raw MOS?)

On Figure 2 it would be very interesting to see how the aggregation of the results from the Alero/CCLM models perform. I would suggest to aggregate the model results in the same way as the observations have been manipulated. It would give an indication of the accuracy of the spatial structure in the model.

When doing comparisons as shown in Figures 5 and 6 the impact of using raw MOS is very clear, leading to very abrubt changes and high noise. Smoothening by fitting a POT distribution to the MOS would lead to results that are easier to interpret. Perhaps showing both types of results in the same figure would lead to an interesting discussion about signal to noise and variation. In any case I find the variation shown in Figure 5 substantial and is less certain about the added benefit of applying non-hydrostatic than your text indicates.

Below are a few detailed comments:

P2, L5. I think you mean 'spatial resolutions up to' rather than 'spatial resolutions down to'

P2, L12: Mayer et al (2015) missing from reference list.

P4, L2: It is difficult to do decadal statistics with only 9 years of simulations.

---

## Referee Comment (RC2) · J. Olsson (Referee) · 14 Apr 2016

The authors compare climate model simulations and observations at different spatial and temporal resolutions with respect to extreme precipitation (rainfall) statistics. Further, (Delta Change) climate factors are calculated and compared for different model resolutions. The topic is interesting and relevant, the material/methods/results are overall accurate as far as I can judge and overall the study should be worth publishing. However, I am not completely satisfied with the presentation and I also have some doubts about the experimental set-up as well as the interpretation of the results; a major revision is recommended.

General comments:

- Using data from just one climate model grid cell is questionable (or, essentially, not allowed), especially when the topic is rainfall extremes. There can be a quite pronounced variability between neighbouring cells and this variability needs to be sampled, maybe by using (at least) 3×3 matrices or something.

- In some figures (e.g. 1, 3, 4 and 6) the authors lump (or pool) observations and simulations on widely different spatial resolutions and sometimes different temporal resolutions too. This first of all makes the figures difficult to read but also the interpretation is rather confusing. Extremes on different resolution conceivably represent different types of physical processes, in different seasons, but this is not much considered in the discussion. The issue is (according to the title) "local impact analysis" and it is not very clear what the low-resolution analyses add in this respect.

- There are different types of observations/analyses (gauge, E-OBS, ERA) as well as many model versions (resolutions, model types, forcing) included in the diagrams but the significance of these different dimensions are seldom assessed but the versions are lumped which makes the text hard to follow. If including all these data/dimensions the results must be accompanied by a very systematic evaluation.

- Please make the figures a bit more reader-friendly by a more systematic use of colours and symbols to represent different dimensions of the data shown (Fig. 6 is quite good in this respect, although it pools different temporal resolutions).

- About permitting convection or not, we have shown that only by increasing the spatial resolution a non-convection permitting model can quite well reproduce local sub-daily extremes (Olsson et al., 2015). Possibly the spatial resolution itself (and associated effects such as topographical representation) is at least (or more) important than whether convection is parameterized or explicit. This needs to be discussed. And it would be interesting to compare both options on the same resolution (maybe already done?), I have heard that in some cases too much convection becomes permitted and the extremes go wild.
I started commenting with a high level of detail below but after the end of page 5 I ran a bit out of time and after that only some selected comments are included.

Specific comments (page, line):

- (1,17-19): Do you intend to say that the high-resolution models better capture local sub-daily extremes than the larger-scale forcing?

- Fig. 1: It would be easier to read if different panels were used for different resolutions (or resolution intervals). Try to use similar symbols or colours to represent similar features in the data (resolution, model, etc.).

- (4,26-30): How well do the different data sets (point, E-OBS) agree on daily/monthly/annual scales? This should be shown.

- (5,6-8): How do you mean "higher resolution results in more extreme precipitation"? Extremes exist at all resolutions.

- (5,9-10): Which 2.8 km CCLM are you referring to here? At least one of them looks quite biased.

- (5,10): "increasing skill with increasing resolution", how do you conclude that from Fig. 1?

- (5,9-15): It is hard to see any clear difference above and below T=2.

- (5,12-13): Very difficult to judge from the figure.

- In Fig. 1 the period 1961-2000 looks to have higher observed daily summer extremes than 2001 but in Fig.2 it looks like the opposite.

- Fig. 2 is basically redundant if dividing Fig. 1 into resolution-specific panels.

- (5,18-20): In winter CNRM-CM3 is closer.

- (5,22-24): Confusing sentence. In what sense does CCLM show a "great ability"? And there is no large underestimation in EC-EARTH what I can see.

[Figure]

- (5,24-25): From the figure it is not obvious that the % bias decreases with increasing time scale.

- Fig. 3: Please add minute and hour on x-axis for improved readability. And I do not like that identical lines and symbols are used to represent different return periods, it is not very helpful for the reader.

- (5,30-31: Which "ALARO runs"? There are several and they are very different, it is not meaningful to talk about "most runs" (same goes for next sentence). What do you mean by extrapolated?

- (5,33): Why "design storms"?

- (6,2-3): I think EC-EARTH agrees quite well with the gridded observations ($\geq$1 day).

- (6,5-7): This sweeping statement about "the CCLM model" is not very helpful; there are many and very different models.

- Fig. 4: What is the added value compared with Fig. 3? How much in Fig. 4 is not based on extremes from JJA. An in-depth look at this issue could potentially reveal interesting features and limitations in the models.

- (6,26): "imaginary extending" is generally not a very accurate concept and quite difficult in this specific case, then better parameterize the curve and extrapolate it.

- (6,35-37): What do you intend to say with this sentence?

- (7,8-9): In relation to what? It would be very surprising if EC-EARTH captured the local sub-daily extremes (also a bit worrying, as it is not supposed to do that). And if imaginary extending the gridded curves the underestimation does not look very remarkable (if I imagine correctly). Please clarify.

- (7,26): It is not relevant to talk about "drier summer" and "wetter winter" when extremes are analysed and not seasonal totals.
- (8,2): Please discuss the figures one at a time, now it is unclear to which figure the following text refers.

- (8,4-7): How do you interpret the fact that summer extremes decrease in the 7-km projections (esp. at 3-h scale)?

- (8,25-27): Do you mean local sub-daily precipitation?

- (8,27-28): Or that the impact of spatial averaging decreases with increasing resolution.

- (9,3-5): Again, do you mean local sub-daily precipitation?

- (9,8-12): Long and hard-to-read sentence.

- 9,15-19): On the daily scale also CCLM(2.8) is quite similar to the driving GCM (EC-EARTH) (Fig. 6), the agreement looks overall similar to ALARO/CNRM (Fig. 5). The differences found seem to be a function of time scale rather than model. Again, this pooling of resolutions makes interpretation difficult.

Olsson, J., Berg, P., and A. Kawamura (2015) Impact of RCM spatial resolution on the reproduction of local, sub-daily precipitation, J. Hydrometeorol., 16, 534–547 doi:10.1175/JHM-D-14-0007

---

## Editor Comment (EC1) · Dr ten Veldhuis (Editor) · 17 Apr 2016

This paper compares climate model simulations and observations at different spatial and temporal resolutions with respect to extreme rainfall statistics. The authors develop climate factors and compare these for different model resolutions. It specifically addresses the gap in knowledge as to which spatio-temporal scale dynamic downscaling should be performed.

The paper was reviewed by 2 reviewers, both experts in the topic covered by the paper. Based on their reviews, I recommend a major revision of the paper, addressing the questions and concerns expressed by the reviewers.

[Figure]

In particular, a more rigorous description of the methodology is required, justifying the choices made in climate modelling approach and metrics chosen for comparing results. The presentation of results is sometimes confusing; especially, it seems that results are compared across different resolutions without explicitly addressing how scale differences influence the results. Also, I would recommend to more explicitly address the role of convection when discussing the modelling results at high resolution: how does the effect of modelling at smaller scales in and of itself relate the impact of including convection, especially when looking at extremes? Overall, the paper would benefit from a more critical discussion of results in chapter 4 and 5. A few specific remarks: - P5, line 5: "good accuracy of the simulations": there is however quite a wide range between point observation and 27.8 km grid values, esp for higher return periods – I would not say there is sufficient agreement to call this "good accuracy" - P5, line 6: "systematic underestimation: this is clearly due to the spatial scale – you should make explicit that you're comparing values for 2 so different scales. - P5, line 9: " nearly unbiased": i.e. unbiased compared to what? Not compared to the station observations: there seems to be a huge difference, esp. for T>5 yrs? - P5, line 16: "difference between climate model outputs and observations may be partly attributed to the spatial scale difference". Exactly, see earlier remark. I suggest you try to explicitly distinguish between differences attributable to spatial scale and to convection permitting model - P5, line 30: "most of the ALARO runs underestimate the station observations". Again, this is likely to be due to the difference in spatial scale - P6, 4-5: "more accurate simulations of 4 convection-permitting models". Is the "higher accuracy", i.e. higher estimated precipitation intensities, due to disaggregating spatial scale or explicitly due to inclusion of convection? This has not been demonstrated in the paper so far. - P7, line 34-35: "Fig. 6 shows change factors for daily and 3-hourly precipitation computed using the CCLMEC-EARTH model with 34 different spatial resolutions for winter and summer seasons. The change factors for all extreme events with T > 1 35 year are shown in this figure." This is not entirely clear to me, better to try and draw a more explicit conclusion: do regional convection-permitting models perform better or Belgium

or not? For what spatial/temporal scales do they perform better? What explains their better performance (just scale or is convection explicitly found to make a difference). Compare IDF values more directly to show % deviations (log- graphs are not very clear to see differences)

---

## Author Comment (AC1) · 12 Jun 2016

Editor: Prof. ten Veldhuis

This paper compares climate model simulations and observations at different spatial and temporal resolutions with respect to extreme rainfall statistics. The authors develop climate factors and compare these for different model resolutions. It specifically addresses the gap in knowledge as to which spatio-temporal scale dynamic downscaling should be performed. The paper was reviewed by 2 reviewers, both experts in the topic covered by the paper. Based on their reviews, I recommend a major revision of the

paper, addressing the questions and concerns expressed by the reviewers. In particular, a more rigorous description of the methodology is required, justifying the choices made in climate modelling approach and metrics chosen for comparing results. The presentation of results is sometimes confusing; especially, it seems that results are compared across different resolutions without explicitly addressing how scale differences influence the results. Also, I would recommend to more explicitly address the role of convection when discussing the modelling results at high resolution: how does the effect of modelling at smaller scales in and of itself relate the impact of including convection, especially when looking at extremes? Overall, the paper would benefit from a more critical discussion of results in chapter 4 and 5.

Response: As suggested by the editor, more details will be added to the methodology section. In addition, the validation and future change sections (Sections 4 and 5) will be revised. The validation section is modified by applying areal reduction factors and adding uncertainty bounds to the at-site observations. The future change section is revised by fitting distribution functions to the change factor curves, and adding a signal-to-noise discussion. Furthermore, the role of convection and spatial scale will be discussed in Sections 4 and 5. We moreover will further improve the readability of the plots.

A few specific remarks: - P5, line 5: "good accuracy of the simulations": there is however quite a wide range between point observation and 27.8 km grid values, esp for higher return periods – I would not say there is sufficient agreement to call this "good accuracy". - P5, line 6: "systematic underestimation: this is clearly due to the spatial scale – you should make explicit that you're comparing values for 2 so different scales. - P5, line 9: " nearly unbiased": i.e. unbiased compared to what? Not compared to the station observations: there seems to be a huge difference, esp. for T>5 yrs? - P5, line 16: "difference between climate model outputs and observations may be partly attributed to the spatial scale difference". Exactly, see earlier remark. I suggest you try to explicitly distinguish between differences attributable to spatial scale and to

convection permitting model. - P5, line 30: "most of the ALARO runs underestimate the station observations". Again, this is likely to be due to the difference in spatial scale. - P6, 4-5: "more accurate simulations of 4 convection-permitting models". Is the "higher accuracy", i.e. higher estimated precipitation intensities, due to disaggregating spatial scale or explicitly due to inclusion of convection? This has not been demonstrated in the paper so far. - P7, line 34-35: "Fig. 6 shows change factors for daily and 3-hourly precipitation computed using the CCLMEC-EARTH model with 34 different spatial resolutions for winter and summer seasons. The change factors for all extreme events with T > 1 35 year are shown in this figure." This is not entirely clear to me, better to try and draw a more explicit conclusion: do regional convection-permitting models perform better in Belgium or not? For what spatial/temporal scales do they perform better? What explains their better performance (just scale or is convection explicitly found to make a difference). Compare IDF values more directly to show % deviations (log- graphs are not very clear to see differences).

Response: We thank the editor for these valuable comments. It should be mentioned that the main focus of this paper is that whether the change factors for precipitation extremes are scale dependent or not. The analysis on the absolute match between climate model outputs and observations was more aimed as an intro to that analysis, to investigate whether higher spatial resolution models provide higher accuracy for precipitation simulations. The aims of the study will be better clarified in the revised paper to meet the above-mentioned comments. Regarding the upscaling of the climate model results, it will be done for checking the accuracy of the spatial structure in the climate models. However, it should be kept in mind that fine-scale data are needed for climate change impact analysis in urban hydrology and these data have to come from the available climate model runs. This is the reason why the focus of the current paper is on the analysis on whether the change factors for precipitation extremes are scale dependent or not. As said, let us clarify this better in the revised manuscript.

================================================================

Referee 1: Prof. Arnbjerg-Nielsen

Overall the paper is well written and shows that the authors have a comprehensive overview of the relevant literature. Further, the work is a logical extension of the work we have done together. The study fits well within the special issue to which it is submitted. It specifically addresses the gap in knowledge on to which spatio-temporal scale dynamic downscaling should be performed. So I I think the work is novel and should be published. I have however some comments that I would like the authors to address that I think will lead to an improved paper.

1- It is probably too late to change now, but I am surprised that you use the A1B scenario for the climate projections. The simulations seem to have been made specifically for this study and hence I had expected an RCP-formulation of the projections.

Response: The climate projections with the ALARO model have been performed a few years ago according to the A1B scenario, and it is still the current scenario used at the Royal Meteorological Institute of Belgium (RMI). However, the ALARO model calculations using the RCP emission pathways are ongoing, and these will be used in future studies.

2- The CCLM model formulation with three nestings seems very complicated. I have no experience with triple nesting and would like to see some references to work indicating that this is a feasible approach. Also a few words on the approach would be nice. Do you apply some sort of nudging or are you only providing boundary conditions? I would have preferred to go directly from the 25*25 km to the highest resolution and then let computational cost define the area. In any case it is unclear whether the 7*7 km is non-hydrostatic or not. So I would appreciate more information on this crucial step.

Response: The integration scale of global models largely differs from convection permitting scale (CPS). A multiple nesting strategy is therefore required to carry out such simulations. Additional nesting steps increase the computational cost and may also add bias at CPS. In this paper, we don't apply the nudging but provide boundary con-

ditions from global model to CCLM for 25 km nest. The 25 km nest was further used to provide boundary conditions to 7 km nest and so on. Brisson et al. (2015) examined the impact of different multiple nesting on the performance of different nesting strategies. They found significant dry bias by removing the 25 km nest. By removing the 25 km and 7 km resolution nests, they found significant model deficiencies. In this case, large-scale precipitation is not well represented and convection hardly occurs resulting in strong underestimation of precipitation. They also found that the impact of 7 km nests is less and may be removed to decrease computational cost. However, it was not fully clear whether this low sensitivity for the 7 km nest is also valid for other meteorological variables and other model configurations. Keeping this uncertainty in view, they used three step nesting in their recent paper (Brisson et al., 2016). We also followed Brisson et al. (2016). It is to be noted that the 7 km run is non-hydrostatic in this case.

Brisson, E., Demuzere, M. and van Lipzig N. P. M.: Modelling strategies for performing convective permitting climate simulations, Meteorologische Zeitschrift, doi: 10.1127/metz/2015/0598, 2015. Brisson E., Van Weverberg, K., M. Demuzere, Devis, A., Saeed, S., Stengel, M., and van Lipzig N. P.M.: How well can a convection-permitting climate model reproduce decadal statistics 2 of precipitation, temperature and cloud characteristics, Climate Dynamics, doi: 10.1007/s00382-016-3012-z, 2016.

3- On page 5 and 6 you have a quite detailed interpretation of how the various statistics perform. It would be nice to have a metric of uncertainty to distinguish between sampling errors and actual signals from the simulations. The most simple would be to include the at-site confidence intervals for the point observations. Another solution could be to consider the variation in the extreme statistics caused by observed decadal oscillations.

Response: We are thankful to the reviewer for this valuable suggestion. The at-site confidence intervals for the point observations will be added to the validation plots in the revised paper.

4- It is not clear to me exactly how the data points in the figures are derived. In Figures 1 and 3 it seems that the points are the raw model output statistics (MOS) plotted using a plotting position formula (which one?). This ignores the dependency of extremes on spatial scale often described by an Areal Reduction Factor (ARF). Plotting these values on the same graph implies that the numbers can be compared directly, which is not the case. Please clarify what you do and consider to modify the graphs by incorporating ARFs.

Response: We plotted raw model output statistics (MOS) using the plotting position formula i/n for given sample size n and i = 1, ..., n. As suggested by the reviewer, Areal Reduction Factor (ARF) will be applied on the climate model simulations to make them directly comparable with the point observations. ARF is computed using the E-OBS data and the point observations for different spatial scales.

5- In Figure 1 I miss an explanation on how you can show 30 year return periods when using 9 years of observations for the CCLM model (it looks like raw MOS?).

Response: Indeed, there was a mistake in the plotting. We also noticed that in the meantime, and would like to apologize for that. In Figure 1, the simulations for the CCLM model should start from the return period of 10 years as we have a 10-year period for the CCLM model (2001-2010). This is now corrected.

6- On Figure 2 it would be very interesting to see how the aggregation of the results from the ALARO/CCLM models perform. I would suggest to aggregate the model results in the same way as the observations have been manipulated. It would give an indication of the accuracy of the spatial structure in the model.

Response: As suggested by the reviewer, the outputs of the ALARO model (and CLLM) will be aggregated to larger pixel sizes and the results will be compared at the same resolution in the revised paper.

7- When doing comparisons as shown in Figures 5 and 6 the impact of using raw MOS

is very clear, leading to very abrupt changes and high noise. Smoothening by fitting a POT distribution to the MOS would lead to results that are easier to interpret. Perhaps showing both types of results in the same figure would lead to an interesting discussion about signal to noise and variation. In any case I find the variation shown in Figure 5 substantial and is less certain about the added benefit of applying non-hydrostatic than your text indicates.

Response: As suggested by the reviewer, smoothening will be done by fitting a distribution to the change factors and a discussion on signal to noise will be presented.

Below are a few detailed comments: P2, L5. I think you mean 'spatial resolutions up to' rather than 'spatial resolutions down to'.

Response: It is corrected.

P2, L12: Mayer et al (2015) missing from reference list.

Response: The reference is added to the reference list.

P4, L2: It is difficult to do decadal statistics with only 9 years of simulations.

Response: We agree with the reviewer. The sentence is revised.

=================================================================
Referee 2: Prof. Olsson

The authors compare climate model simulations and observations at different spatial and temporal resolutions with respect to extreme precipitation (rainfall) statistics. Further, (Delta Change) climate factors are calculated and compared for different model resolutions. The topic is interesting and relevant, the material/methods/results are overall accurate as far as I can judge and overall the study should be worth publishing. However, I am not completely satisfied with the presentation and I also have some doubts about the experimental set-up as well as the interpretation of the results; a major revision is recommended.

- Using data from just one climate model grid cell is questionable (or, essentially, not allowed), especially when the topic is rainfall extremes. There can be a quite pronounced variability between neighbouring cells and this variability needs to be sampled, maybe by using (at least) 3×3 matrices or something.

Response: As suggested by the reviewer, the climate data for a matrix of 3 × 3 (9) model grid points surrounding the closest model grid point to Uccle will also be analyzed in the revised paper.

- In some figures (e.g. 1, 3, 4 and 6) the authors lump (or pool) observations and simulations on widely different spatial resolutions and sometimes different temporal resolutions too. This first of all makes the figures difficult to read but also the interpretation is rather confusing. Extremes on different resolution conceivably represent different types of physical processes, in different seasons, but this is not much considered in the discussion. The issue is (according to the title) "local impact analysis" and it is not very clear what the low-resolution analyses add in this respect. - There are different types of observations/analyses (gauge, E-OBS, ERA) as well as many model versions (resolutions, model types, forcing) included in the diagrams but the significance of these different dimensions are seldom assessed but the versions are lumped which makes the text hard to follow. If including all these data/dimensions the results must be accompanied by a very systematic evaluation.

Response: The figures are revised for better interpretation of the results. In addition, E-OBS data are removed from the figures after applying areal reduction factor (suggested by the reviewer 1) on the climate model outputs.

- Please make the figures a bit more reader-friendly by a more systematic use of colours and symbols to represent different dimensions of the data shown (Fig. 6 is quite good in this respect, although it pools different temporal resolutions).

Response: The colors and symbols in the plots are revised as per suggestion.

- About permitting convection or not, we have shown that only by increasing the spatial resolution a non-convection permitting model can quite well reproduce local sub-daily extremes (Olsson et al., 2015). Possibly the spatial resolution itself (and associated effects such as topographical representation) is at least (or more) important than whether convection is parameterized or explicit. This needs to be discussed. And it would be interesting to compare both options on the same resolution (maybe already done?), I have heard that in some cases too much convection becomes permitted and the extremes go wild.

Response: The results of the climate models will be compared at the same spatial scale and a discussion about the role of spatial scale and convection on the results of ALARO and CCLM model will be added to the revised paper.

I started commenting with a high level of detail below but after the end of page 5 I ran a bit out of time and after that only some selected comments are included. Specific comments (page, line): - (1,17-19): Do you intend to say that the high-resolution models better capture local sub-daily extremes than the larger-scale forcing?

Response: Revision is done as per suggestion.

- Fig. 1: It would be easier to read if different panels were used for different resolutions (or resolution intervals). Try to use similar symbols or colours to represent similar features in the data (resolution, model, etc.).

Response: The symbols in the plots are revised.

- (4,26-30): How well do the different data sets (point, E-OBS) agree on daily/monthly/annual scales? This should be shown.

Response: A comparison will be done between point and gridded precipitation for different time scales.

- (5,6-8): How do you mean "higher resolution results in more extreme precipitation"? Extremes exist at all resolutions.

**HESSD**

Response: We agree with the reviewer that extremes exist at all resolutions, but climate models with higher resolutions present more extreme precipitation.

- (5,9-10): Which 2.8 km CCLM are you referring to here? At least one of them looks quite biased. - (5,10): "increasing skill with increasing resolution", how do you conclude that from Fig. 1?

Response: We will modify the climate model output validation and related discussion in the text.

- (5,9-15): It is hard to see any clear difference above and below T=2.

Response: The paragraph is modified.

- (5,12-13): Very difficult to judge from the figure.

Response: The figure is revised.

- In Fig. 1 the period 1961-2000 looks to have higher observed daily summer extremes than 2001 but in Fig.2 it looks like the opposite.

Response: There is an error in plotting the CCLM results. The precipitation quantiles for the CCLM model should start from 10-year return period (for a 10-year period of 2001-2010). It is also noted that the precipitation extremes with T >1 year were averaged for Figure 2.

- Fig. 2 is basically redundant if dividing Fig. 1 into resolution-specific panels.

Response: Fig. 1 shows only validation of daily precipitation quantiles, while Fig. 2 shows the validation for the daily, weekly and monthly time scales and for averaged extreme values.

- (5,18-20): In winter CNRM-CM3 is closer.

Response: Yes, CNRM-CM3 is closer to the observations in winter.

- (5,22-24): Confusing sentence. In what sense does CCLM show a "great ability"?

And there is no large underestimation in EC-EARTH what I can see.

Response: The statement for the CCLM is modified. We also agree with the reviewer that underestimation in EC-EARTH is not large. The sentence is revised.

- (5,24-25): From the figure it is not obvious that the % bias decreases with increasing time scale.

Response: Percentage bias is not shown in Figure 2. The sentence is based on un-shown results.

- Fig. 3: Please add minute and hour on x-axis for improved readability. And I do not like that identical lines and symbols are used to represent different return periods, it is not very helpful for the reader.

Response: Minute, hour and day will be added to IDF plots. Different symbols and lines will be used for different return periods to enhance readability.

- (5,30-31): Which "ALARO runs"? There are several and they are very different, it is not meaningful to talk about "most runs" (same goes for next sentence). What do you mean by extrapolated?

Response: The sentences are modified. The extrapolation is done for sub-daily data.

- (5,33): Why "design storms"?

Response: The sentence is revised.

- (6,2-3): I think EC-EARTH agrees quite well with the gridded observations ($\geq$ 1 day).

Response: Large underestimation of EC-EARTH GCM is for sub-daily precipitation. For larger time scales, its performance is quite well.

- (6,5-7): This sweeping statement about "the CCLM model" is not very helpful; there are many and very different models.

Response: The statement is modified.

- Fig. 4: What is the added value compared with Fig. 3? How much in Fig. 4 is not based on extremes from JJA. An in-depth look at this issue could potentially reveal interesting features and limitations in the models.

Response: We agree with the reviewer that most of the extremes in Fig.4 are based on extremes from JJA which are shown in Fig. 3. So, we keep Fig. 3 which includes some sub-daily ALARO runs and remove Fig. 4.

- (6,26): "imaginary extending" is generally not a very accurate concept and quite difficult in this specific case, then better parameterize the curve and extrapolate it.

Response: The extrapolation is done as per suggestion.

- (6,35-37): What do you intend to say with this sentence?

Response: After applying ARF, the gridded data will be removed from the plot and therefore this sentence is removed from the text.

- (7,8-9): In relation to what? It would be very surprising if EC-EARTH captured the local sub-daily extremes (also a bit worrying, as it is not supposed to do that). And if imaginary extending the gridded curves the underestimation does not look very remarkable (if I imagine correctly). Please clarify.

Response: ARF will be applied on the EC-EARTH output and the comparison will be made in relation to point observations. The extrapolation will also be done based on the previous suggestion by the reviewer.

- (7,26): It is not relevant to talk about "drier summer" and "wetter winter" when extremes are analysed and not seasonal totals.

Response: We agree with the reviewer and the sentences are corrected.

(8,2): Please discuss the figures one at a time, now it is unclear to which figure the following text refers.

[Figure]

Response: The figures are cited in several places in the text to show to which figures the text refers.

- (8,4-7): How do you interpret the fact that summer extremes decrease in the 7-km projections (esp. at 3-h scale)?

Response: A distribution will be fitted to the change factor curve of the 7-km model (as suggested by reviewer 1) and then the obtained changes relative to noise will be discussed.

- (8,25-27): Do you mean local sub-daily precipitation?

Response: Revision is done as per suggestion.

- (8,27-28): Or that the impact of spatial averaging decreases with increasing resolution.

Response: This issue is investigated by comparing the model results at the same resolution (aggregation of the model results at finer scale to larger scale).

- (9,3-5): Again, do you mean local sub-daily precipitation?

Response: Revision is done as per suggestion.

- (9,8-12): Long and hard-to-read sentence.

Response: The sentence is revised.

- (9,15-19): On the daily scale also CCLM(2.8) is quite similar to the driving GCM (ECEARTH) (Fig. 6), the agreement looks overall similar to ALARO/CNRM (Fig. 5). The differences found seem to be a function of time scale rather than model. Again, this pooling of resolutions makes interpretation difficult.

Response: We agree with the reviewer that at the daily scale the results for the two convection-permitting models are similar. Since we did not analyze changes in sub-daily precipitation from ALARO model, it can be said that the difference is a function of

time scale. The sentence is revised.

---

## Author Response (AR1)

**Editor: Prof. ten Veldhuis**

This paper compares climate model simulations and observations at different spatial and temporal resolutions with respect to extreme rainfall statistics. The authors develop climate factors and compare these for different model resolutions. It specifically addresses the gap in knowledge as to which spatio-temporal scale dynamic downscaling should be performed.

The paper was reviewed by 2 reviewers, both experts in the topic covered by the paper. Based on their reviews, I recommend a major revision of the paper, addressing the questions and concerns expressed by the reviewers.

In particular, a more rigorous description of the methodology is required, justifying the choices made in climate modelling approach and metrics chosen for comparing results. The presentation of results is sometimes confusing; especially, it seems that results are compared across different resolutions without explicitly addressing how scale differences influence the results. Also, I would recommend to more explicitly address the role of convection when discussing the modelling results at high resolution: how does the effect of modelling at smaller scales in and of itself relate the impact of including convection, especially when looking at extremes? Overall, the paper would benefit from a more critical discussion of results in chapter 4 and 5.

*Response: As suggested by the editor, more details were added to the methodology section including the procedure used for selection of independent extreme values based on the POT method and the method used for the change factor calculations after extreme value distribution fitting. In addition, the validation and future change sections (Sections 4 and 5) were revised completely based on the reviewers' comments. The validation section was modified by adding uncertainty bounds to the at-site observations and comparing the model results at the same spatial resolution to exclude the effect of the spatial scale difference on the results. Some preliminary analyses were also added based on the reviewers' comments which include i) a comparison between the climate model results for a matrix of $3 \times 3$ model grid points surrounding the closest model grid point to Uccle station and ii) a comparison between the point and pixel observations for different time scales and seasons. The future change section was revised by fitting distribution functions to the change factor curves, and the extreme value distribution based change factors were compared with those from the empirical data. We moreover improved the readability of the plots.*

A few specific remarks:

- P5, line 5: "good accuracy of the simulations": there is however quite a wide range between point observation and 27.8 km grid values, esp for higher return periods – I would not say there is sufficient agreement to call this "good accuracy".

- P5, line 6: "systematic underestimation: this is clearly due to the spatial scale – you should make explicit that you're comparing values for 2 so different scales.

- P5, line 9: " nearly unbiased": i.e. unbiased compared to what? Not compared to the station observations: there seems to be a huge difference, esp. for T>5 yrs?

- P5, line 16: "difference between climate model outputs and observations may be partly attributed to the spatial scale difference". Exactly, see earlier remark. I suggest you try to explicitly distinguish between differences attributable to spatial scale and to convection permitting model.

- P5, line 30: "most of the ALARO runs underestimate the station observations". Again, this is likely to be due to the difference in spatial scale.

- P6, 4-5: "more accurate simulations of 4 convection-permitting models". Is the "higher accuracy", i.e. higher estimated precipitation intensities, due to disaggregating spatial scale or explicitly due to inclusion of convection? This has not been demonstrated in the paper so far.

- P7, line 34-35: "Fig. 6 shows change factors for daily and 3-hourly precipitation computed using the CCLMEC-EARTH model with different spatial resolutions for winter and summer seasons. The change factors for all extreme events with T > 1 year are shown in this figure." This is not entirely clear to me, better to try and draw a more explicit conclusion: do regional convection-permitting models perform better in Belgium or not? For what spatial/temporal scales do they perform better? What explains their better performance (just scale or is convection explicitly found to make a difference). Compare IDF values more directly to show % deviations (log- graphs are not very clear to see differences).

*Response: We thank the editor for these valuable comments.*

*It should be mentioned that the main focus of this paper is that whether the change factors for precipitation extremes are scale dependent or not. The analysis on the absolute match between climate model outputs and observations was more aimed as an intro to that analysis, to investigate whether higher spatial resolution models provide higher accuracy for precipitation simulations.*

*Regarding the upscaling of the climate model results, it will be done for checking the accuracy of the spatial structure in the climate models. However, it should be kept in mind that fine-scale data are needed for climate change impact analysis in urban hydrology and these data have to come from the available climate model runs. This is the reason why the focus of the current paper is on the analysis on whether the change factors for precipitation extremes are scale dependent or not.*

==================================================================================

**Referee 1: Prof. Arnbjerg-Nielsen**

Overall the paper is well written and shows that the authors have a comprehensive overview of the relevant literature. Further, the work is a logical extension of the work we have done together. The study fits well within the special issue to which it is submitted.

It specifically addresses the gap in knowledge on to which spatio-temporal scale dynamic downscaling should be performed. So l I think the work is novel and should be published. I have however some comments that I would like the authors to address that I think will lead to an improved paper.

1- It is probably too late to change now, but I am surprised that you use the A1B scenario for the climate projections. The simulations seem to have been made specifically for this study and hence I had expected an RCP-formulation of the projections.

*Response: The climate projections with the ALARO model have been performed a few years ago according to the A1B scenario, and it is still the current scenario used at the Royal Meteorological Institute of Belgium (RMI). However, the ALARO model calculations using the RCP emission pathways are ongoing, and these will be used in future studies.*

2- The CCLM model formulation with three nestings seems very complicated. I have no experience with triple nesting and would like to see some references to work indicating that this is a feasible approach. Also a few words on the approach would be nice. Do you apply some sort of nudging or are you only providing boundary conditions? I would have preferred to go directly from the 25*25 km to the highest resolution and then let computational cost define the area. In any case it is unclear whether the 7*7 km is non-hydrostatic or not. So I would appreciate more information on this crucial step.

*Response: The integration scale of global models largely differs from convection permitting scale (CPS). A multiple nesting strategy is therefore required to carry out such simulations. Additional nesting steps increase the computational cost and may also add bias at CPS. In this paper, we don't apply the nudging but provide boundary conditions from global model to CCLM for 25 km nest. The 25 km nest was further used to provide boundary conditions to 7 km nest and so on.*

*Brisson et al. (2015) examined the impact of different multiple nesting on the performance of different nesting strategies. They found significant dry bias by removing the 25 km nest. By removing the 25 km and 7 km resolution nests, they found significant model deficiencies. In this case, large-scale precipitation is not well represented and convection hardly occurs resulting in strong underestimation of precipitation. They also found that the impact of 7 km nests is less and may be removed to decrease computational cost. However, it was not fully clear whether this low sensitivity for the 7 km nest is also valid for other meteorological variables and other model configurations. Keeping this uncertainty in view, they used three step nesting in their recent paper (Brisson et al., 2016). We also followed Brisson et al. (2016). It is to be noted that the 7 km run is non-hydrostatic in this case.*

*Brisson, E., Demuzere, M. and van Lipzig N. P. M.: Modelling strategies for performing convective permitting climate simulations, Meteorologische Zeitschrift, doi: 10.1127/metz/2015/0598, 2015.*

*Brisson E., Van Weverberg, K., M. Demuzere, Devis, A., Saeed, S., Stengel, M., and van Lipzig N. P.M.: How well can a convection-permitting climate model reproduce decadal statistics of precipitation, temperature and cloud characteristics, Climate Dynamics, doi: 10.1007/s00382-016-3012-z, 2016.*

3- On page 5 and 6 you have a quite detailed interpretation of how the various statistics perform. It would be nice to have a metric of uncertainty to distinguish between sampling errors and actual signals from the simulations. The most simple would be to include the at-site confidence intervals for the point observations. Another solution could be to consider the variation in the extreme statistics caused by observed decadal oscillations.

*Response: We are thankful to the reviewer for this valuable suggestion. The at-site confidence intervals for the point observations were added to the validation plots in the revised paper (see Figures 3 and 4).*

4- It is not clear to me exactly how the data points in the figures are derived. In Figures 1 and 3 it seems that the points are the raw model output statistics (MOS) plotted using a plotting position formula (which one?). This ignores the dependency of extremes on spatial scale often described by an Areal Reduction Factor (ARF). Plotting these values on the same graph implies that the numbers can be compared directly, which is not the case. Please clarify what you do and consider to modify the graphs by incorporating ARFs.

*Response: We plotted raw model output statistics (MOS) using the plotting position formula i/n for given sample size n and i = 1, ..., n.*

*As suggested by the reviewer, Areal Reduction Factors (ARFs) were calculated using the E-OBS data and the point observations for different periods and seasons. As one can see from the plots below (Figure L1), there is a large uncertainty in ARFs. We afraid that applying these ARFs would add more errors to the analysis rather than an added value. Hence, we didn't apply these ARFs to the climate model outputs and the results were discussed compared to the point and gridded observations in the revised paper.*

[Figure]

***Figure L1*** *Areal reduction factors for different spatial resolutions computed using the E-OBS data and the point observations for different periods and seasons.*

5- In Figure 1 I miss an explanation on how you can show 30 year return periods when using 9 years of observations for the CCLM model (it looks like raw MOS?)

*Response: Indeed, there was a mistake in the plotting. We also noticed that in the meantime, and would like to apologize for that. In Figure 4, the simulations for the CCLM model should start from the return period of 10 years as we have a 10-year period for the CCLM model (2001-2010). This is now corrected.*

6- On Figure 2 it would be very interesting to see how the aggregation of the results from the ALARO/CCLM models perform. I would suggest to aggregate the model results in the same way as the observations have been manipulated. It would give an indication of the accuracy of the spatial structure in the model.

*Response: As suggested by the reviewer, the outputs of the ALARO model (and CLLM) were aggregated to larger pixel sizes and the results were compared at the same resolution in the revised paper (see Figures 3 and 4 and also the discussed results in Section 4 in the revised paper).*

7- When doing comparisons as shown in Figures 5 and 6 the impact of using raw MOS is very clear, leading to very abrupt changes and high noise. Smoothening by fitting a POT distribution to the MOS would lead to results that are easier to interpret. Perhaps showing both types of results in the same figure would lead to an interesting discussion about signal to noise and variation. In any case I find the variation shown in Figure 5 substantial and is less certain about the added benefit of applying non-hydrostatic than your text indicates.

*Response: As suggested by the reviewer, smoothening was done by fitting an extreme value distribution to the change factors (see Figure 7 and 8 in the revised paper). The empirical data based and the extreme value distribution based change factors were discussed in Section 5.*

Below are a few detailed comments:

P2, L5. I think you mean 'spatial resolutions up to' rather than 'spatial resolutions down to'.

*Response: It was corrected.*

P2, L12: Mayer et al (2015) missing from reference list.

*Response: The reference was added to the reference list.*

P4, L2: It is difficult to do decadal statistics with only 9 years of simulations.

*Response: We agree with the reviewer. The sentence was revised.*

==========================================================================

**Referee 2: Prof. Olsson**

The authors compare climate model simulations and observations at different spatial and temporal resolutions with respect to extreme precipitation (rainfall) statistics. Further, (Delta Change) climate factors are calculated and compared for different model resolutions. The topic is interesting and relevant, the material/methods/results are overall accurate as far as I can judge and overall the study should be worth publishing.

However, I am not completely satisfied with the presentation and I also have some doubts about the experimental set-up as well as the interpretation of the results; a major revision is recommended.

- Using data from just one climate model grid cell is questionable (or, essentially, not allowed), especially when the topic is rainfall extremes. There can be a quite pronounced variability between neighbouring cells and this variability needs to be sampled, maybe by using (at least) 3×3 matrices or something.

*Response: As suggested by the reviewer, the climate data for a matrix of 3 × 3 (9) model grid points surrounding the closest model grid point to Uccle were analyzed in the revised paper. The analysis for the ALARO model was shown*

*in Figures L2, L3, L4 and L5. As shown, the extreme precipitation intensity in the pixel covering Uccle station is within the value range provided by the surrounding pixels. The analysis for hourly precipitation extremes in a matrix of 3×3 ALARO$_{ERA-Interim}$ model was included as an example in the revised paper (see Figure 1 in the revised paper).*

[Figure]

[Figure]

*Figure L2 Comparison of daily precipitation quantiles in a matrix of 3×3 ALARO model grid points surrounding the closest model grid point to Uccle (Gridcell 5), for summer season.*

[Figure]

[Figure]

***Figure L3*** *Comparison of hourly precipitation quantiles in a matrix of 3×3 ALARO model grid points surrounding the closest model grid point to Uccle (Gridcell 5), for summer season.*

[Figure]

***Figure L4*** *Comparison of daily precipitation quantiles in a matrix of 3×3 ALARO 4 km model grid points surrounding the closest model grid point to Uccle (Gridcell 5), for winter season.*

[Figure]

***Figure L5*** *Comparison of hourly precipitation quantiles in a matrix of 3×3 ALARO 4 km model grid points surrounding the closest model grid point to Uccle (Gridcell 5), for winter season.*

- In some figures (e.g. 1, 3, 4 and 6) the authors lump (or pool) observations and simulations on widely different spatial resolutions and sometimes different temporal resolutions too. This first of all makes the figures difficult to read but also the interpretation is rather confusing. Extremes on different resolution conceivably represent different types of physical processes, in different seasons, but this is not much considered in the discussion. The issue is (according to the title) "local impact analysis" and it is not very clear what the low-resolution analyses add in this respect.

- There are different types of observations/analyses (gauge, E-OBS, ERA) as well as many model versions (resolutions, model types, forcing) included in the diagrams but the significance of these different dimensions are

seldom assessed but the versions are lumped which makes the text hard to follow. If including all these data/dimensions the results must be accompanied by a very systematic evaluation.

*Response: The figures were revised for a better interpretation of the results. We used different colors and symbols in the plots for a better presentation of the results. It is worthwhile to note that all the climate model outputs are presented next to each other on one plot for a better comparison of the results (to compare the model outputs of different resolutions, and also the driven models with driving GCM/reanalysis and to investigate the effect of boundary condition on the model results).*

- Please make the figures a bit more reader-friendly by a more systematic use of colours and symbols to represent different dimensions of the data shown (Fig. 6 is quite good in this respect, although it pools different temporal resolutions).

*Response: The colors and symbols in the plots were revised as per suggestion.*

- About permitting convection or not, we have shown that only by increasing the spatial resolution a non-convection permitting model can quite well reproduce local sub-daily extremes (Olsson et al., 2015). Possibly the spatial resolution itself (and associated effects such as topographical representation) is at least (or more) important than whether convection is parameterized or explicit. This needs to be discussed. And it would be interesting to compare both options on the same resolution (maybe already done?), I have heard that in some cases too much convection becomes permitted and the extremes go wild.

*Response: The results of the climate models were compared at the same spatial scale (see Figures 3 and 4 in the revised paper) and the results for the native and aggregated quantiles were discussed in Section 4 in the revised paper.*

I started commenting with a high level of detail below but after the end of page 5 I ran a bit out of time and after that only some selected comments are included.

Specific comments (page, line):

- (1,17-19): Do you intend to say that the high-resolution models better capture local sub-daily extremes than the larger-scale forcing?

*Response: Revision was done as per suggestion.*

- Fig. 1: It would be easier to read if different panels were used for different resolutions (or resolution intervals). Try to use similar symbols or colours to represent similar features in the data (resolution, model, etc.).

*Response: The symbols in the plots were revised in all the figures. Different panels for different resolutions makes it difficult to compare the models with different resolutions.*

- (4,26-30): How well do the different data sets (point, E-OBS) agree on daily/monthly/annual scales? This should be shown.

*Response: As suggested by the reviewer, a comparison was done between the point and gridded precipitation for different time scales (see Figure 2 in the revised paper).*

- (5,6-8): How do you mean "higher resolution results in more extreme precipitation"? Extremes exist at all resolutions.

*Response: We agree with the reviewer that extremes exist at all resolutions, but climate models with higher resolutions present more extreme precipitation.*

- (5,9-10): Which 2.8 km CCLM are you referring to here? At least one of them looks quite biased.
- (5,10): "increasing skill with increasing resolution", how do you conclude that from Fig. 1?

*Response: We modified the climate model output validation and related discussion in the text (see Section 4).*

- (5,9-15): It is hard to see any clear difference above and below T=2.

*Response: The paragraph was modified.*

- (5,12-13): Very difficult to judge from the figure.

*Response: The figure was revised.*

- In Fig. 1 the period 1961-2000 looks to have higher observed daily summer extremes than 2001 but in Fig.2 it looks like the opposite.

*Response: There was an error in plotting the CCLM results. The precipitation quantiles for the CCLM model should start from 10-year return period (for a 10-year period of 2001-2010). It is also noted that the precipitation extremes with T >1 year were averaged for Figure 4.*

- Fig. 2 is basically redundant if dividing Fig. 1 into resolution-specific panels.

*Response: Figure 1 (Figures 3 and 4 in the revised paper) shows only validation of daily precipitation quantiles for summer, while Figure 2 (Figures 5 in the revised paper) shows the validation for the daily, weekly and monthly time scales and for averaged extreme values of the summer and winter seasons.*

- (5,18-20): In winter CNRM-CM3 is closer.

*Response: Yes, CNRM-CM3 is closer to the observations in winter.*

- (5,22-24): Confusing sentence. In what sense does CCLM show a "great ability"?
And there is no large underestimation in EC-EARTH what I can see.

*Response: The statement for the CCLM was modified.*

*The underestimation in EC-EARTH is quiet large. Please note that the EC-EARTH results should be compared with the dotted line which corresponds to the period 2001-2010.*

- (5,24-25): From the figure it is not obvious that the % bias decreases with increasing time scale.

*Response: Percentage bias is not shown in the mentioned figure (Figure 5 in the revised paper). The sentence is based on unshown results.*

- Fig. 3: Please add minute and hour on x-axis for improved readability. And I do not like that identical lines and symbols are used to represent different return periods, it is not very helpful for the reader.

*Response: Minute, hour and day were added to the IDF plots. Moreover, different symbols and lines were used for different return periods to enhance readability in Figure 6 of the revised paper.*

- (5,30-31): Which "ALARO runs"? There are several and they are very different, it is not meaningful to talk about "most runs" (same goes for next sentence). What do you mean by extrapolated?

*Response: The sentences were modified. The extrapolation was done for sub-daily data.*

- (5,33): Why "design storms"?

*Response: The sentence was revised.*

- (6,2-3): I think EC-EARTH agrees quite well with the gridded observations (≥ 1 day).

*Response: Large underestimation of EC-EARTH GCM is for sub-daily precipitation. For larger time scales, its performance is quite well.*

- (6,5-7): This sweeping statement about "the CCLM model" is not very helpful; there are many and very different models.

*Response: The statement was modified.*

- Fig. 4: What is the added value compared with Fig. 3? How much in Fig. 4 is not based on extremes from JJA. An in-depth look at this issue could potentially reveal interesting features and limitations in the models.

*Response: We agree with the reviewer that most of the extremes in Figure 4 are based on extremes from JJA which are shown in Figure 3. So, we kept Figure 3 (Figure 6 in the revised paper) which includes some sub-daily ALARO runs and removed Fig. 4.*

- (6,26): "imaginary extending" is generally not a very accurate concept and quite difficult in this specific case, then better parameterize the curve and extrapolate it.

*Response: The extrapolation for sub-daily precipitation intensity of the E-OBS data was done based on extreme value distribution fitting (see Figure 6).*

- (6,35-37): What do you intend to say with this sentence?

*Response: From a physical point of view, the underestimation of the climate models may be partially due to spatial scale difference or because of the deficiency of the model itself. Whatever the reason of this underestimation is from a physical point of view, in practice the IDF curves are constructed based on the raingauge precipitation since hydrologists believe that the raingauge precipitation data are the most accurate data.*
*The sentence was removed from the revised paper.*

- (7,8-9): In relation to what? It would be very surprising if EC-EARTH captured the local sub-daily extremes (also a bit worrying, as it is not supposed to do that). And if imaginary extending the gridded curves the underestimation does not look very remarkable (if I imagine correctly). Please clarify.

*Response: The extrapolation was done based on extreme value distributions. As one can see, the underestimation of sub-daily precipitation intensity from the gridded observations by the EC-EARTH GCM is remarkable for 10-year return period.*

- (7,26): It is not relevant to talk about "drier summer" and "wetter winter" when extremes are analysed and not seasonal totals.

*Response: We agree with the reviewer and the sentences were corrected.*

(8,2): Please discuss the figures one at a time, now it is unclear to which figure the following text refers.

*Response: The figures were cited in several places in the text to show to which figures the text refers.*

- (8,4-7): How do you interpret the fact that summer extremes decrease in the 7-km projections (esp. at 3-h scale)?

*Response: A distribution was fitted to the change factor curve of the 7-km model (as suggested by reviewer 1) and then the difference between the changes based on the extreme value distribution and the empirical data was discussed.*

- (8,25-27): Do you mean local sub-daily precipitation?

*Response: Revision was done as per suggestion.*

- (8,27-28): Or that the impact of spatial averaging decreases with increasing resolution.

*Response: This issue was investigated by comparing the model results at the same resolution (aggregation of the model results at finer scale to larger scale).*

- (9,3-5): Again, do you mean local sub-daily precipitation?

*Response: Revision was done as per suggestion.*

- (9,8-12): Long and hard-to-read sentence.

*Response: The sentence was revised.*

- (9,15-19): On the daily scale also CCLM(2.8) is quite similar to the driving GCM (ECEARTH) (Fig. 6), the agreement looks overall similar to ALARO/CNRM (Fig. 5). The differences found seem to be a function of time scale rather than model. Again, this pooling of resolutions makes interpretation difficult.

*Response: We agree with the reviewer that at the daily scale the results for the two convection-permitting models are similar. However, there is an amplification of the changes in 
[revised manuscript text omitted]

---

## Author Response (AR2)

**Referee 1: Prof. Arnbjerg-Nielsen**

The revision has led to a substantially improved paper. Two minor things should be considered.

1- The new sentence in the abstract is very complex - in fact I have read it several times and still do not understand it. Please divide into several sentences.

*Response: The comment was applied accordingly in the abstract.*

2- Also, the sentence \*A multiple nesting strategy is therefore required to carry out such simulations\* should be modified. In the rebuttal letter you mention one group of people making one study. That hardly justifies this generic statement. I suggest to mention the references you have cited in the rebuttal letter and say that you have therefore chosen the approach that you use.

*Response: The sentence was revised as per suggestion.*

[revised manuscript text omitted]